# Contribution of oxic methane production to surface methane emission in lakes and its global importance

Marco Günthel[1]*, Daphne Donis [2], Georgiy Kirillin[3], Danny Ionescu [4], Mina Bizic[4], Daniel F. McGinnis [2]*, Hans-Peter Grossart [4,5]* & Kam W. Tang[1]*

Recent discovery of oxic methane production in sea and lake waters, as well as wetlands, demands re-thinking of the global methane cycle and re-assessment of the contribution of oxic waters to atmospheric methane emission. Here we analysed system-wide sources and sinks of surface-water methane in a temperate lake. Using a mass balance analysis, we show that internal methane production in well-oxygenated surface water is an important source for surface-water methane during the stratified period. Combining our results and literature reports, oxic methane contribution to emission follows a predictive function of littoral sediment area and surface mixed layer volume. The contribution of oxic methane source(s) is predicted to increase with lake size, accounting for the majority (>50%) of surface methane emission for lakes with surface areas >1 km$^2$.

[1] Department of Biosciences, Swansea University, SA2 8PP Swansea, UK. [2] Aquatic Physics Group, Department F.-A. Forel for Environmental and Aquatic Sciences (DEFSE), Faculty of Science, University of Geneva, 1211 Geneva, Switzerland. [3] Department of Ecohydrology, Leibniz Institute of Freshwater Ecology and Inland Fisheries, 12587 Berlin, Germany. [4] Department of Experimental Limnology, Leibniz Institute of Freshwater Ecology and Inland Fisheries, 16775 Stechlin, Germany. [5] Institute of Biochemistry and Biology, Potsdam University, 14476 Potsdam, Germany. *email: marcoguenthel@gmail.com; daniel.mcginnis@unige.ch; hgrossart@igb-berlin.de; k.w.tang@swansea.ac.uk

After carbon dioxide, methane is the second most important carbon-based greenhouse gas[1,2], and its continuous increase in the atmosphere is a global climate threat[3,4]. A basic premise in methane biogeochemistry is that biological methane formation occurs exclusively under anoxic conditions[5,6]. Over the past several decades[7] there have been multiple reports of paradoxical methane oversaturation in oxic sea and lake waters (Tang et al.[8] and references herein). This methane paradox can be resolved by attributing the methane to conventional anoxic sources[9,10], or by additionally considering oxic–water methane production (OMP). The idea of OMP goes against the long-standing paradigm in methane research, and despite the skepticism[11,12], different investigators have confirmed repeatedly that methane production can and does occur under oxic condition in sea and lake waters[13–16], and studies have begun to identify the responsible organisms[17–19] and the underlying biochemical pathways[17,20]. Unlike anoxic methane sources in sediments and bottom waters, methane production in the surface-mixed layer (SML) places the methane source closer to the water–air interface, and therefore its contribution to surface emission can be significant[8,21].

Globally, it is estimated that freshwaters account for (mean ± minimum error range) $122 \pm 60 \, \mathrm{Tg \, yr^{-1}}$ methane to the atmosphere (ca. 20% of the total emission)[22]. However, this emission value is not well constrained as indicated by the large uncertainty range[22], and leads to disagreement between bottom-up and top-down methane budgets[22,23]. The large uncertainty of freshwater emission during upscaling is commonly attributed to highly variable methane density fluxes within and across systems[24–27], scarcity of long-term data, which do not cover high ecosystem variability[22,28], and uncertainties in global freshwater areas[29–31]. Oxic methane production (OMP) has so far not been considered in global assessments, including methane budgets[22,23] and IPCC reports[1,2] despite its potential to contribute significantly to methane density fluxes in freshwater systems[15,21,32]. For more accurate modeling of freshwater emission and corresponding contribution to the global methane budget, a better understanding of internal methane production, consumption, and distribution pathways is needed.

While methanogenic Archaea are largely responsible for anoxic methane production[6,33], primary production has been associated with the oxic methane source[15,17,32,34]. Therefore, the oxic and anoxic sources will react differently to environmental factors. Global methane budget assessments and future climate change predictions will benefit from proper distinction of oxic versus anoxic methane sources and identifying their individual contribution to the system-wide emission. Bogard et al.[32] conducted experiments in Lake Cromwell (Canada) and estimated that OMP accounted for 20% of the total surface emission, with the rest originating from anoxic sources. Likewise, Donis et al.[21] estimated that OMP was the main methane source in the SML of Lake Hallwil (Switzerland) and accounted for 63–83% of the surface emission (value updated, see Supplementary Note 1 including Supplementary Fig. 1, Supplementary Tables 1 and 2). While both studies demonstrate that OMP can be an important source of methane emission, it is not clear if OMP is a general phenomenon in lakes and what may explain the different contribution patterns in different lakes.

Unlike the open ocean, oxic methane production in lake waters can be confounded by anoxic methane input from the littoral zone. To resolve this, we conducted a study in Lake Stechlin where we used experimental enclosures (Leibniz-Institute of Freshwater Ecology and Inland Fisheries, The Lake Lab; https://www.lake-lab.de (2012); Supplementary Fig. 2a) to examine the lake-water methane dynamics without the influence from the littoral zone. Lake Stechlin is a medium-size ($4.25 \, \mathrm{km^2}$) meso-oligotrophic lake with a mean depth of 22.7 m (max. 69.5 m) in Northeastern Germany (Supplementary Fig. 2b). It has negligible river in-/outflow, small groundwater-feed[35] and has been monitored for decades by the Leibniz Institute for Freshwater Ecology and Inland Fisheries (IGB)[36]. The Lake Lab installed in Lake Stechlin's South basin consists of a series of experimental enclosures (with periodic water exchange) and a central reservoir (no water exchange since installation in 2011/2012). Methane oversaturation in the lake's surface oxic layer has been observed since 2010[15,16,34]. Throughout the years 2014, 2016, and 2018 we measured dissolved methane concentration, surface methane emission, and environmental parameters (temperature, dissolved oxygen, algal pigments, and wind speed) in the Northeast and South basins and inside the enclosures (see Supplementary Table 3 for data overview). We then used the data to conduct a detailed methane mass balance analysis for the SML, accounting for the different sources and sinks (Fig. 1), including lateral methane input and OMP under different seasonal conditions (mixed and stratified seasons), and compared our mass balance results to earlier findings. Finally, we combined our findings with literature data to develop a predictive model for oxic methane contribution in relation to lake morphology, and discussed its significance in the global context. Our results show that the contribution of oxic methane source to lake surface emission increases with lake size. Accordingly, in lakes larger than $1 \, \mathrm{km^2}$ (or with a littoral sediment area to SML volume ratio smaller than $0.07 \, \mathrm{m^2 \, m^{-3}}$) the oxic source dominates methane surface emission. Applying the predictive model to the global lake inventory ($\geq 0.01 \, \mathrm{km^2}$) shows that oxic methane production may account for up to 66% of lake methane emission worldwide. This finding highlights that future assessments of global methane emissions should include oxic methane source(s) and that more research is needed to understand the impact of oxic methane production in various lake types and its responses to environmental perturbation such as global warming and widespread eutrophication.

## Results

**Environmental condition**. Temperature and buoyancy frequency $N^2$ profiles indicate that Lake Stechlin was completely mixed in 2016 until April (Supplementary Fig. 3). At the end of April 2016, the lake started to warm and thermal stratification was established during May. From June to August, the lake was clearly stratified with temperatures $\geq 20 \, ^\circ\mathrm{C}$ in the SML. As the stratified water column was mainly sampled during June and July, we refer to this period as the stratified period unless stated otherwise. The thickness of the SML was about 5 m during June, and 6 m in July and August.

Throughout the study period, the water column never turned anoxic, with dissolved oxygen reaching up to ca. $17 \, \mathrm{mg \, l^{-1}}$ (ca. 170% saturation) typically 1 m below the methane peak (Supplementary Fig. 4).

**Methane concentration**. With the onset of stratification, methane concentrations in the oxic upper water column in both Northeast and South basins increased sharply, reaching up to $1400 \, \mathrm{nmol \, l^{-1}}$ at thermocline depth (6 m). The SML remained oversaturated with methane throughout the stratified season in both basins ($400–900 \, \mathrm{nmol \, l^{-1}}$), while methane concentrations were less than $200 \, \mathrm{nmol^{-1} \, l}$ at >10 m depth (Fig. 2a, b).

Inside the experimental enclosures (water exchanged with open-lake water 2 weeks prior to sampling), methane concentrations were also at over-saturation level in the SML ($300–400 \, \mathrm{nmol \, l^{-1}}$) with a profile similar to the open water, except for a smaller methane peak at the thermocline (Fig. 2c, d). In contrast, the central reservoir (water never exchanged since installation in 2011/2012) showed a

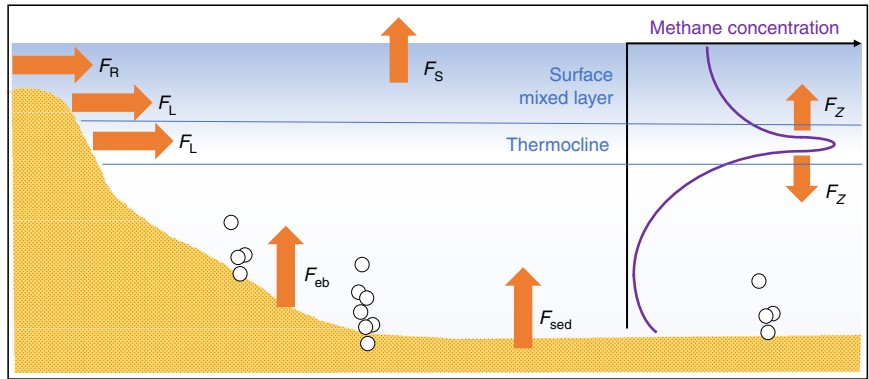

**Fig. 1** Methane fluxes in lakes. The typical methane profile of the lake water column has a distinct peak within the thermocline. Methane is introduced into the surface mixed layer horizontally by lateral transport from peripheral water bodies ($F_R$) and littoral sediments ($F_L$) and vertically via (turbulent) diffusion ($F_z$) originating from bottom sediments (ebullitive flux $F_{eb}$, diffusive flux $F_{sed}$). Methane is released to the atmosphere ($F_S$) across the water–air interface. Biological modulation accounts for additional methane sink and source. Methane loss due to oxidation by methanotrophs is commonly acknowledged, whereas oxic methane production in the surface mixed layer represents an overlooked part of the global methane cycle (e.g., IPCC 2007[1] and IPCC 2013[2]) (picture drafted as after Donis et al.[21]).

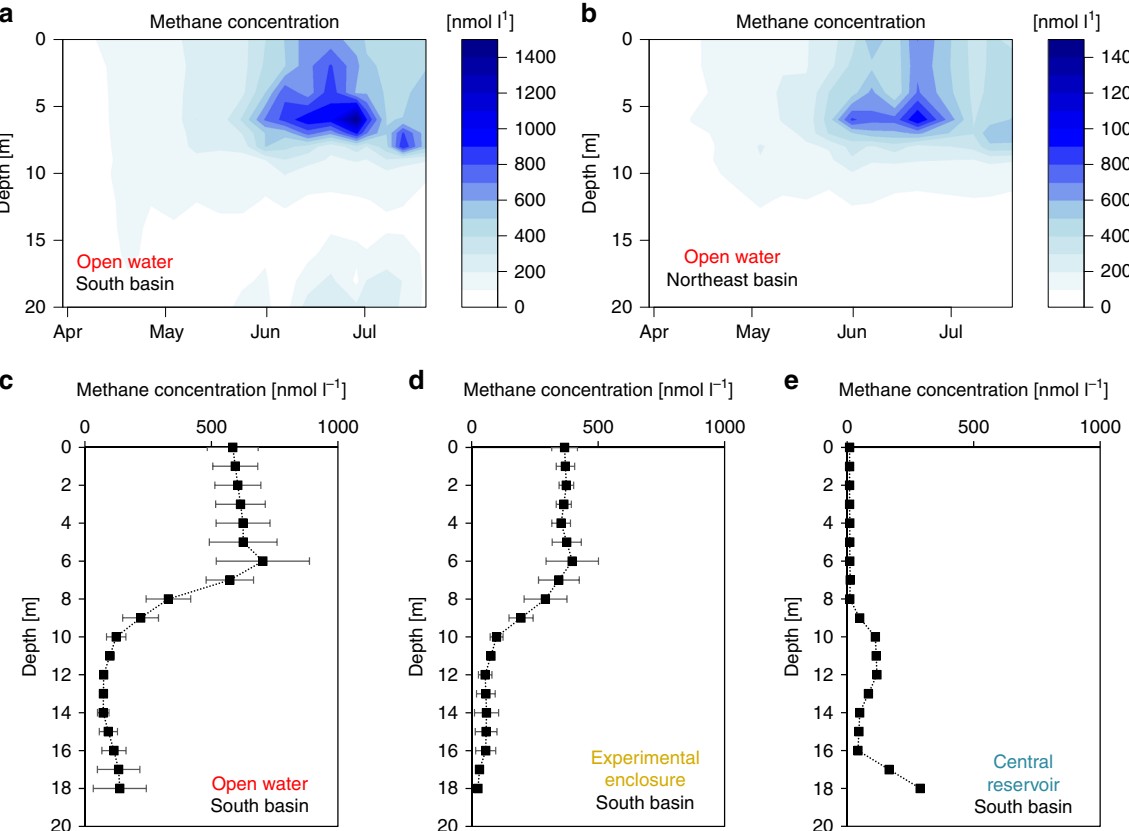

**Fig. 2** Methane accumulation in the water column. Panel **a** shows the in situ methane concentration [nmol l$^{-1}$] recorded weekly in 2016 in the South basin (53°08'36.6''N 13°01'42.8''E). Increasing concentration indicates accumulation. Panel **b** shows the methane concentration [nmol l$^{-1}$] recorded weekly in 2016 in the Northeast basin (53°09'20.2''N 13°01'51.5''E). Note, panel **a** contains an additional data point compared to panel **b** in the end of June. Panel **c** shows the methane profile in the open lake of the South basin (53°08'36.6''N 13°01'42.8''E; 20.5 m deep) as mean ± SD of 4 profiles taken on 4 different days in August 2014. Panel **d** shows the methane profile inside experimental enclosure 1 (53°08'36.4''N 13°01'41.6''E; ca. 20 m deep) as mean ± SD of 4 profiles taken on 4 different days in August 2014. Panel **e** illustrates the methane profile inside the central enclosure (53°08'35.8''N 13°01'41.1''E; ca. 18.5 m deep) as mean ± SD of methodological duplicate measurement taken on 7$^{th}$ July 2016. Source data are provided as a Source Data file.

completely different profile during the stratified period, with negligible amount of methane in the SML (≤15 nmol l$^{-1}$) and higher concentration of methane below 16 m (300 nmol l$^{-1}$) (Fig. 2e). The small peak (120 nmol l$^{-1}$) at 12 m depth in the central reservoir methane profile appears to be the result of a

different methane production–consumption balance at this depth, but has not been examined in detail.

**Surface methane emission.** The surface methane emission ($F_S$) was either measured using a flux chamber (all Northeast basin

values except on 20[th] June) or estimated from a wind-based model (all other values) that was developed from the flux chamber measurements and concurrent wind conditions. Emission data were transformed to gas transfer constants $k_{600}$ as a linear function of wind speed ($U_{10}$, recorded at 10 m height), $k_{600}$ [cm h$^{-1}$] = 1.98 * $U_{10}$ [m s$^{-1}$] + 0.94 ($R^2 = 0.44$, $p < 0.01$). This linear function was then used to estimate surface emissions in the South basin (enclosures and open lake) based on wind speed (Supplementary Note 2, Supplementary Table 4). Other published models[21,37,38] in the literature (mainly based on direct turbulence measurements)[37,38] were used to validate these emission values (see sensitivity analysis in Discussion).

In the Northeast basin the surface methane emission increased by an order of magnitude from the non-stratified period (March: mean ± SD; 0.049 ± 0.026 mmol m$^{-2}$ d$^{-1}$) to the stratified period (0.47 ± 0.27 mmol m$^{-2}$ d$^{-1}$). Compared to the Northeast basin, higher surface emission was observed in the South basin during the stratified period (mean ± SD; 0.71 ± 0.24 mmol m$^{-2}$ d$^{-1}$).

The experimental enclosures showed a surface methane flux of (mean ± SD) 0.43 ± 0.07 mmol m$^{-2}$ d$^{-1}$ in August 2014, which was about half of the flux measured in the adjacent open water (0.77 mmol m$^{-2}$ d$^{-1}$) at the same time. In contrast, the central reservoir showed a much lower surface methane emission of 0.01 mmol m$^{-2}$ d$^{-1}$ (measurement on 7[th] July). Details on flux parametrization are summarized in Supplementary Note 2.

**Vertical methane diffusion**. Diffusivity ($K_z$) was high in the SML, but it decreased to ca. 10$^{-6}$ m$^2$ s$^{-1}$ at the upper boundary of the thermocline in the stratified period (Supplementary Fig. 3c). Consequently, the diffusive methane input from the thermocline to the SML ($F_z$) during the stratified season was small for both the Northeast: (mean ± SD) 0.032 ± 0.031 mmol m$^{-2}$ d$^{-1}$ and the South basin: 0.050 ± 0.065 mmol m$^{-2}$ d$^{-1}$, and negligible in the central reservoir (4.4 × 10$^{-4}$ mmol m$^{-2}$ d$^{-1}$).

When the diffusive methane input was compared between experimental enclosures and open water in August 2014, the experimental enclosures showed lower values (mean ± SD; 0.007 ± 0.009 mmol m$^{-2}$ d$^{-1}$) than the adjacent open water (0.024 mmol m$^{-2}$ d$^{-1}$).

**Lateral input from littoral zones**. Methane measurements were done in the experimental enclosures and the adjacent open water (South basin) in August 2014. The experimental enclosures were shielded from the littoral zone (e.g., no lateral methane input), therefore OMP in the SML was estimated from Eq. (1) (see Method section) without the $F_L$ term to be (mean ± SD) 101 ± 17 nmol l$^{-1}$ d$^{-1}$ (Supplementary Table 5). By comparing the data from the experimental enclosures and those from the adjacent open water (both collected in the South basin) and deploying mass balance, we estimated the transport of methane from littoral sediments within the SML to the lake pelagic water to be 76 ± 12 nmol l$^{-1}$ d$^{-1}$ (Supplementary Table 5), which corresponds to an average littoral sediment methane flux ($F_L$) of (mean ± SD) 1.4 ± 0.2 mmol m$^{-2}$ d$^{-1}$.

**Oxic methane production**. OMP at high temporal resolution (approximately weekly) in the two open-water sites was estimated from Eq. (1) (see Method section) using as $F_L$ term (lateral methane input) the value obtained for August 2014 as described above. During the non-stratified season, OMP rates were negligible and then slowly increased in late April/May 2016 (Fig. 3). As the water column became fully stratified, the average OMP rate between the two basins ranged between 26 and 236 nmol l$^{-1}$ d$^{-1}$, reaching the maximum for both basins (259 nmol l$^{-1}$ d$^{-1}$ in

Northeast basin, 214 nmol l$^{-1}$ d$^{-1}$ in South basin) in late June (Fig. 3).

Monte Carlo simulation was applied to assess uncertainties in the mass balance for the stratified period, and the resultant OMP rates in the SML were (mean ± SD) 72 ± 74 nmol l$^{-1}$ d$^{-1}$ (84% probability of positive value) for the Northeast basin and 88 ± 75 nmol l$^{-1}$ d$^{-1}$ for the South basin (Table 1). On average, OMP contributed 64% of the surface methane emission in the Northeast basin, and 50% in the South basin, with the remaining methane originating from anoxic sources. A sensitivity analysis (see discussion) examined the effect of variable mass balance components on the contribution pattern.

**Predicting oxic methane contribution from lake morphology**. Our analysis shows that lateral input from the littoral zone and in situ OMP were the two major SML methane sources, together accounting for ≥95% of the surface emission in Lake Stechlin. While the estimated OMP rate was comparable between the two basins, its relative importance, expressed as the percentage of oxic methane contribution to the system-wide emission (*OMC*), was considerably higher in the Northeast basin than in the South basin. This difference was explained by the difference in geo-morphology between the two basins: lateral input is a function of littoral sediment area ($A_{sed}$), whereas OMP is a function of the volume of SML across the lake basin (∀). The relative importance between lateral input versus in situ OMP is therefore scaled to $A_{sed}$/∀, which decreases with increasing basin size.

While Stechlin's Northeast and South basins vary in surface area (NE: 2.01 km$^2$; S: 1.12 km$^2$) and SML volume ∀ (NE: 11,200,000 m$^3$; S: 5,700,000 m$^3$), their littoral sediment areas are comparable (NE: 0.28 km$^2$, S: 0.31 km$^2$) (values given for a 6 m deep SML). As expected, *OMC* was higher in the larger Northeast basin (64%) compared to the smaller South basin (50%) due to a smaller $A_{sed}$/∀ ratio in the Northeast basin.

We extended this scaling exercise to other temperate oligo- to mesotrophic lakes of various sizes extracted from the literature[21,32,39] (Supplementary Note 3, Supplementary Table 6) in order to derive an empirical relationship between *OMC* and lake morphology. The data showed that *OMC* is a negative log-linear function of $A_{sed}$/∀ (Fig. 4). Least square regression after linearization gave a highly significant $p$ value (≪0.01) and a high $R^2$ value (0.95). A significant relationship was also found between *OMC* and lake surface area (Supplementary Fig. 6). Both functions predicted that the importance of OMP for SML methane increases with lake size; for lakes with $A_{sed}$/∀ ≤ 0.07 m$^2$ m$^{-3}$ or surface area ≥ 1 km$^2$, OMP is expected to be the main source (>50%) of surface methane emissions.

**Discussion**

In this study, we balanced the methane sources in two basins of the temperate meso-oligotrophic Lake Stechlin in high temporal resolution covering the shift from mixed to stratified water column conditions. We further analyzed the methane budget in two different types of enclosures, both isolated from littoral methane input: in experimental enclosures (1200 m$^3$) where water is periodically exchanged (last time 2 weeks prior to sampling) and in the central reservoir (14,000 m$^3$) where water has not been exchanged since installation in 2011/2012 and is likely nutrient depleted. Comparing the methane budgets in the open water and enclosures allowed us to demonstrate that stratification mainly disconnected SML methane from bottom sediment methanogenesis, that OMP occurred irrespective of littoral influence, and that OMP contributed substantially to the system-wide methane emission of Lake Stechlin's Northeast (64%) and South basin (50%) exceeding the littoral methane source contribution (32% in

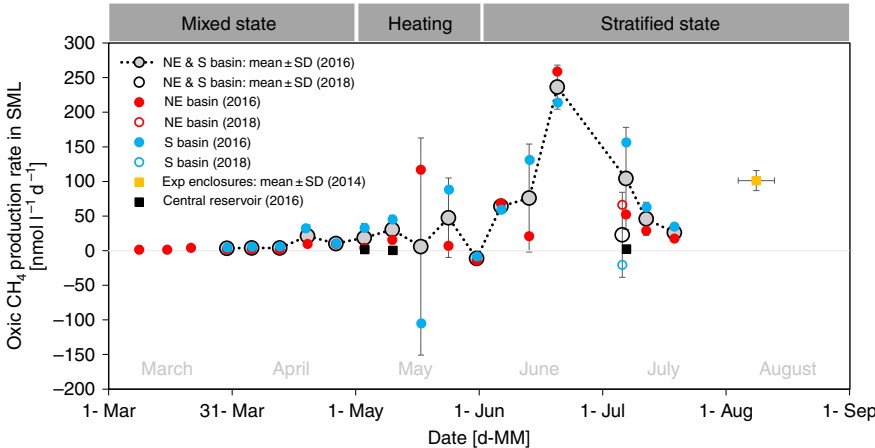

**Fig. 3** Oxic methane production rates. Production rates were computed using a mass balance approach. Red circles represent measurements in the open water of the Northeast basin (69.5 m deep; 53°09'20.2''N 13°01'51.5''E) and blue circles measurements in the open water of the South basin (20.5 m deep; 53°08'36.6''N 13°01'42.8''E). Gray circles are average values of both basins. The yellow square is the average value for the experimental enclosures of the lake lab facility (enclosures 1 and 13), and black squares are measurements in the central reservoir. Vertical error bars illustrate standard deviation from mean values; and horizontal error bars (only experimental enclosures) depict the time frame of corresponding sampling. The mass balance was estimated for unstratified condition in March/April 2016 (negligible lateral methane flux, negligible methane oxidation) and for stratified condition June–August 2014/2016/2018 (lateral methane input from sediments: 1.4 mmol m$^{-2}$ d$^{-1}$; 30% of internally produced methane is oxidation). For May 2016, non-stratified parametrization was used for the first half of the month and stratified parametrization for the second half. Methane surface emission was measured in the Northeast basin (except on 20$^{th}$ June 2016) and on 6$^{th}$ July 2018 in the South basin, and was estimated for the other sites based on wind speed parametrization. The sampling schedule for all field measurements is laid out in Supplementary Table 3. Source data are provided as a Source Data file.

**Table 1 Mass balance components.**

| Site | Mass balance component | Symbol | Whole system | | Per volume |
|---|---|---|---|---|---|
| | | | [mol d$^{-1}$] | [kg d$^{-1}$] | [nmol l$^{-1}$ d$^{-1}$] |
| Northeast basin | Surface emission | $F_S$ | 942 ± 538 | 15 ± 9 | 90 ± 52 |
| | Methane oxidation | $MOx$ | 226 | 4 | 22 |
| | Lateral sediment input | $F_L$ | 372 ± 57 | 6 ± 1 | 36 ± 6 |
| | Diffusion from thermocline | $F_z$ | 56 ± 55 | 1 ± 1 | 5 ± 5 |
| | Internal (oxic) production | $P_{net}$ | 752 ± 771 | 12 ± 12 | **72 ± 74** |
| South basin | Surface emission | $F_S$ | 795 ± 268 | 13 ± 4 | 148 ± 50 |
| | Methane oxidation | $MOx$ | 141 | 2 | 26 |
| | Lateral sediment input | $F_L$ | 423 ± 65 | 7 ± 1 | 79 ± 12 |
| | Diffusion from thermocline | $F_z$ | 41 ± 54 | 1 ± 1 | 8 ± 10 |
| | Internal (oxic) production | $P_{net}$ | 470 ± 400 | 8 ± 6 | **88 ± 75** |

Oxic production was computed by measuring/estimating surface emission, oxidation, lateral input, as well as vertical diffusion (see Fig. 1) and solving the mass balance for the missing component Seven replicate measurements were taken in the open water of the Northeast (69.5 m deep; surface area 2,006,700 m$^2$; 53°09'20.2''N 13°01'51.5''E) and South basin (20.5 m deep; surface area 1,122,775 m$^2$; 53°08'36.6''N 13°01'42.8''E) of Lake Stechlin during the stratified period in 2016 (June–July). Values listed as mean ± SD. Note that Monte Carlo simulation was used to solve the mass balance after the target component (in bold; mean ± 1 SD) (see Methods for details). Supplementary Fig. 5 illustrates the density function of the Northeast and South basin dataset. If the Monte Carlo simulation were to be applied to whole lake data (combining South and Northeast basins data), oxic methane production rates (denoted as $P_{net}$ in Eq. (1)) do not change: 78 ± 80 nmol l$^{-1}$ d$^{-1}$ ($F_S$ = 2503 ± 1160, $MOx$ = 496, $F_L$ = 1198 ± 185, $F_z$ = 139 ± 170, $P_{net}$ = 1653 ±1703 mol d$^{-1}$)

the Northeast basin and 45% in the South basin). Finally, combining mass balance results for Lake Stechlin and literature data for other lakes allowed us to develop a predictive model estimating the contribution of OMP to the system-wide methane surface emission as a function of lake morphological parameters, and the model suggests that OMP has important ramifications especially in large stratified lakes.

Mass balance approach has been successfully used by others to study methane dynamics in lakes[40], including OMP[21,32]. However, this approach is sensitive to the accuracy of the individual components of the mass balance. Therefore, to assess the validity and robustness of our mass balance analysis, we evaluated the different components by comparing our measurements with literature values and examined how variabilities of the mass balance components may alter the overall

conclusion. The average surface methane emission ($F_S$) during the stratified period was 0.47 mmol m$^{-2}$ d$^{-1}$ (±57% SD) in the Northeast basin and 0.71 mmol m$^{-2}$ d$^{-1}$ (±34% SD) in the South basin (taken mainly during calm weather). The larger value in the South basin can be attributed to higher influence from littoral methane sources. However, these emission values are comparable with the global estimate of 0.62 mmol m$^{-2}$ d$^{-1}$ for the region 25–54° latitude[41] and within the range reported earlier for Lake Stechlin[42] (exceeding 4 mmol m$^{-2}$ d$^{-1}$ at strong wind; on average 2.6 mmol m$^{-2}$ d$^{-1}$ ± 42% SD). Highly variable surface emission has been reported earlier, for some systems standard deviations exceed 100% of mean emission values during summer[24,26]. In case of the South basin we estimated the emission from wind speed data and the corresponding results are dependent on the gas transfer constant ($k_{600}$) value used.

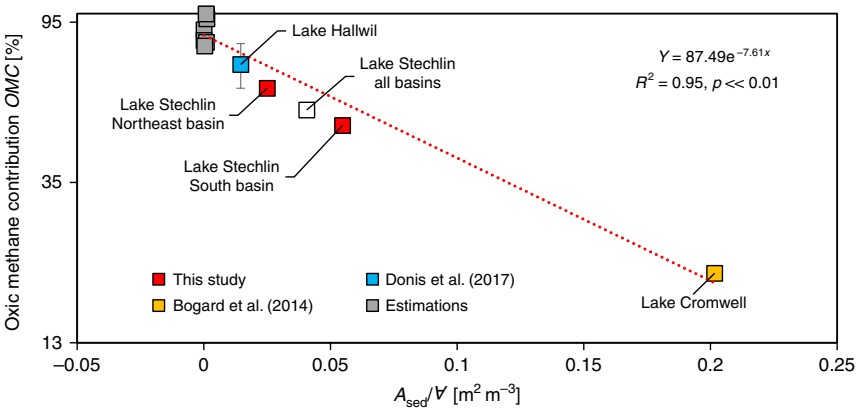

**Fig. 4** Oxic methane contribution versus lake morphology. The ratio of sediment area ($A_{sed}$) and surface mixed layer volume ($\forall$) determines the oxic methane contribution to surface emission (OMC). The trend line (red line) follows the exponential function $y = 87.49e^{-7.61x}$ ($R^2 = 0.95$, $p \ll 0.01$, standard error = 8.6%). The y-axis is scaled to $\log_{2.7}$ and the x-axis is linear. With increasing lake size, $\forall$ increases quicker than $A_{sed}$ making oxic methane production the largest source of surface mixed layer methane in lakes with $A_{sed}/\forall \leq 0.07 \ m^2 \ m^{-3}$. Lake Hallwil estimation[21] was updated as described in Supplementary Note 1; the lower and upper end (error bars) were used to compute the mean OMC which was used for developing the trend line function. Estimations for other lakes were computed as defined in Supplementary Note 3. If whole lake data (combining South and Northeast basin data) was to be applied to this empirical model (empty symbol) the regression constants and statistics only change minimally ($y = 88.48e^{-7.56x}$; $R^2 = 0.96$, $p \ll 0.01$). Source data are provided as a Source Data file.

Our $k_{600}$-wind speed relationship ($k_{600} \ [cm \ h^{-1}] = 1.98 \times U_{10} \ [m \ s^{-1}] + 0.98$) was very similar to an earlier report (e.g., Lake Hallwil: $k_{600} \ [cm \ h^{-1}] = 2.0 \times U_{10} \ [m \ s^{-1}]$; Donis et al.[21]). Applying six alternative emission models (based on wind or combined wind and lake size) presented by Vachon and Prairie[37], MacIntyre et al.[38] and Donis et al.[21] to this dataset resulted in an average emission rate between 0.55 and 1.03 mmol $m^{-2} \ d^{-1}$. Applying these alternative emission rates to the mass balance analysis gave an OMP rate between 41 and 185 nmol $l^{-1} \ d^{-1}$, which still translated to a substantial oxic methane contribution (32–68%) to the surface methane emission (details in Supplementary Table 7). In other words, regardless of the method or model used to estimate surface methane emission, it remains that OMP was an important contributor to surface emission.

Comparing the methane data inside the experimental enclosures with that of the open water gave an average lateral methane input ($F_L$) of 1.4 mmol $m^{-2} \ d^{-1}$ from the littoral sediment. It is within the range of fluxes reported for other temperate water bodies (e.g., Rzeszów Reservoir, Poland[43]: (mean ± SD) 0.69 ± 0.56 mmol $m^{-2} \ d^{-1}$ in May–Sep; Lake Hallwil, Switzerland[21]: 1.75 ± 0.2 mmol $m^{-2} \ d^{-1}$ in Sep (Supplementary Note 1); Boltzmann–Arrhenius equation at ca. 20 °C[12]: ca. 2 mmol $m^{-2} \ d^{-1}$, including Lake Constance (Überlingen basin)/Lake Ammer/Lake Königsegg/Reservoir Schwarzbach in Germany[12] with ca. 1.3 mmol $m^{-2} \ d^{-1}$). Even doubling the lateral methane input, what is an unlikely scenario for a meso-oligotrophic lake such as Lake Stechlin, still could not fully explain the observed SML methane in the Northeast basin, and a substantial OMP rate (19 nmol $l^{-1} \ d^{-1}$) would still be required to balance the methane budget. More importantly, within the experimental enclosures, which were isolated from lateral input, the estimated OMP was (mean ± SD) 101 ± 17 nmol $l^{-1} \ d^{-1}$ (Aug 2014 dataset), which was comparable to the estimated average OMP in the open water for both basins (72–88 nmol $l^{-1} \ d^{-1}$) (June/July 2016 dataset).

The calculation of methane diffusive input from the lower water layers ($F_z$) is dependent on the estimated $K_z$ value (diffusivity). Our $K_z$ values were comparable to an earlier report for the same lake[36]. Even in Lake Hallwil, which is 5–10 times larger than the Lake Stechlin basins and is therefore exposed to stronger seiching effects, very similar $K_z$ values were observed[21]

(thermocline minimum about $10^{-6} \ m^2 \ s^{-1}$). The SML methane in Lake Stechlin was decoupled from bottom sediment methanogenesis during thermal stratification, as it is also indicated by the methane-depth profile of the central reservoir (Fig. 2e) where water has not been exchanged since installation in 2011/2012. Accordingly, methane diffusion from Lake Stechlin's thermocline water accounted for only 2–5% (likely overestimated) of the SML methane in the open-water sites, and only 1% in the experimental enclosures. Variability in the corresponding mass balance components, therefore, was negligible and would not affect the overall conclusion.

The magnitude of methane oxidation (MOx) varies between seasons[44–46] and between lakes[39]. Oxygen concentration[47] and light[48,49] are important modulating factors for MOx in lake surface waters. In other lakes, MOx rates in oxic surface waters have been reported to range between 4 and 30 nmol $l^{-1} \ d^{-1}$ [21,32,50]. For our study, we assumed MOx to be equivalent to a constant fraction (30%) of the internal production during the stratified season (see method section for details). The average OMP rates for both basins were 72–88 nmol $l^{-1} \ d^{-1}$, giving a hypothetical MOx rate of ca. 24 nmol $l^{-1} \ d^{-1}$, which is within the range of literature values. Because methane oxidation is parameterized as a loss term in the mass balance analysis, higher MOx would translate to higher OMP, and vice versa. If we consider the extreme scenario by completely ignoring methane oxidation (MOx = 0), the estimated average OMP rate for the South basin would decrease to (mean ± SD) 40 ± 53 nmol $l^{-1} \ d^{-1}$ and would still remain an important SML methane source (32%).

Comparing our measurements and assumptions against literature values shows that our mass balance analysis is reasonably parametrized and robust. The system-wide methane emission from the SML in the Northeast basin was estimated to be 942 mol $d^{-1}$ in the stratified period, of which 32% from lateral input (372 mol $d^{-1}$) and 5% from vertical diffusion from the thermocline (56 mol $d^{-1}$) (Table 1). Similarly, methane emission from the SML in the South basin was 795 mol $d^{-1}$, and only 45% (423 mol $d^{-1}$) could be attributed to lateral input and 4% (41 mol $d^{-1}$) to vertical input from the thermocline. The deficits (plus additional consumption via methanotrophy), therefore, must be compensated for by internal OMP. The estimated OMP rate averaged over the stratified period was (mean ± SD) 72 ± 74 nmol $l^{-1} \ d^{-1}$ (Northeast basin) and 88 ±

75 nmol l⁻¹ d⁻¹ (South basin). An earlier study[15] using bottle incubations measured a net OMP rate of up to 58 nmol l⁻¹ d⁻¹ for Lake Stechlin, which corresponds to a hypothetical gross production rate of 75 nmol l⁻¹ d⁻¹ when assuming 30% oxidation. Similar OMP rates have also been estimated for Lake Hallwil, between 76 and 138 nmol l⁻¹ d⁻¹ [21] (Supplementary Note 1). Particularly high OMP values, such as what we found in late June (mean ± SD; 236 ± 32 nmol l⁻¹ d⁻¹), have also been reported by others[32] (e.g., 230 ± 10 nmol l⁻¹ d⁻¹ in Lake Cromwell, Canada). Overall, by accounting for the different methane sources and sinks in the SML mass balance analysis, we show that OMP is a key contributor to system-wide surface emission in Lake Stechlin. This conclusion is consistent with previously reported OMP rates obtained from bottle incubations[15] and is not sensitive to inherent uncertainties in our mass balance approach as shown by the sensitivity analysis.

In addition to known knowledge gaps in the global methane dynamics[22,23], OMP has not been considered as source of uncertainty in global assessments[1,2,22,23]. Because both oxic and anoxic methane sources in lakes can be modulated by multiple factors and processes (Supplementary Fig. 7), some of which are still poorly understood, it would be premature to construct a mechanistic model to fully describe methane dynamics in lakes. Instead, we developed empirical models as useful tools to predict the contribution of OMP to the system-wide emission (OMC) in stratified meso-to-oligotrophic lakes in the temperate region based on a set of simple lake morphological parameters (Fig. 4, Supplementary Fig. 6). The first model using littoral sediment area ($A_{sed}$) and SML volume ($\forall$) as proxy explains nearly the entire variance in the dataset ($R^2 = 0.95$, $p \ll 0.01$) making it a powerful predictive model to estimate OMC from $A_{sed}$ and $\forall$. For cases where $A_{sed}$ and $\forall$ data are unavailable, OMC can be related to easily accessible lake surface area (Supplementary Fig. 6). With an average accuracy of 91.4% (standard error = 8.6%) this model also provides reliable OMC estimates. Both empirical models predict the importance of OMP for atmospheric emission to increase with lake size.

The system-wide contribution of the anoxic methane sources is mainly controlled by littoral sediment flux and the corresponding littoral sediment area. Trophic state[51,52] and temperature[12,53] are important drivers of the methane flux from sediments. Higher sediment methane fluxes in eutrophic systems and in warmer climate zones compared to our dataset of stratified meso-to-oligotrophic lakes in the temperate region could shift the curve of the empirical models to the right (Fig. 4, Supplementary Fig. 6). However, sediment methane fluxes vary in a rather narrow range by a factor of 26 between oligotrophic and eutrophic lakes[52] (e.g., 0.2–5.2 mmol m⁻² d⁻¹). Likewise, reported average OMP rates varied by a factor of 6 in stratified lakes[15,21,32] (40–230 nmol l⁻¹ d⁻¹ including this study). In comparison, our predictive model covers lake surface area that varies by a factor of 190,000. The OMC prediction, therefore, may vary mainly for small lakes which have been reported to cause less methane emission on a global scale compared to large lakes[28] (<0.01 versus >1 km²). It shall be noted that the model predictions based on $A_{sed}$ and $\forall$ will be more reliable than based exclusively on lake surface area due to sediment steepness, aspect ratio and total depth modulating the littoral sediment area at constant lake surface area.

Methane emission from lakes has been identified as a key contributor of this powerful greenhouse gas to the atmosphere[22]. It is therefore a legitimate question to ask: how important is OMP in this context on a global scale? To get a first-order estimation, we applied our empirical model to the global lake size distributions based on satellite data, which covers lakes ≥0.01 km²[31]. The result suggests that globally, an average of 66% of lake methane emission may have originated from oxic production

(Supplementary Note 4, Supplementary Table 8). Such a surprising finding justifies the need for further investigation of OMP in lakes worldwide with different geological histories, trophic states, climates, and physical (e.g., lake color, stratification patterns or with strong in-/out flow) and chemical characteristics (e.g., alkaline versus acidic) (Supplementary Fig. 7). By increasing data resolution in our empirical models, the models can then be used to further improve the global methane emission assessments.

Unlike the anoxic methane production driven by anaerobic methanogens with enzymes that are oxygen-sensitive[54], OMP in lake waters has been attributed to novel biochemical pathways involving photoautotrophs[15,34,55]. Our system-wide methane mass balance demonstrates that without OMP a substantial methane source is missing when balancing Lake Stechlin's SML methane sources and sinks. The estimated OMP rates agree very well with earlier results from bottle incubation experiments[15] and account for ≥50% of the system-wide methane emission. Following our model, OMC is predicted to be the major methane source for the system-wide emission in lakes >1 km². In the light of global warming and widespread lake eutrophication, stratification periods will extend[56,57] and phytoplankton production in the SML is expected to increase worldwide[58], which may increase OMP and its contribution to methane emission to the atmosphere. To understand and predict future climate change scenarios, it is crucial to consider lake water OMP in the global methane assessment and how it responds to environmental perturbations.

## Methods

**Study site.** Lake Stechlin (Germany) is a meso-oligotrophic temperate glacial lake. For this study, we focused on the Northeast and South basins. Typical of temperate lakes, the water column of Lake Stechlin is well mixed in winter, begins to stratify in April/May and remains stratified until September or October. Throughout the stratified period, the oxygen-rich SML and thermocline are oversaturated with methane[19,34].

The Lake Lab facility was installed in the South basin in 2011/2012, which consists of 24 experimental enclosures (each 9 m diameter × 20 m depth) and a central reservoir (30 m diameter × 20 m depth), all of which extend into the bottom sediment. Water in the experimental enclosures 1 and 13 of the Lake Lab facility was exchanged with open lake water 2 weeks prior to our study; the water in the central reservoir has never been changed since installation.

Parameters of lake morphology, such as volume of the SML ($\forall$) and planar areas ($A_{tot}$, $A_{th}$, $A_{sed}$), were derived from thermocline depth data and bathymetry data. Supplementary Table 9 summarizes the parameterization of the mass balance for open-water and enclosure calculations for the stratified (June–July 2016/2018; Aug 2014) and the non-stratified periods (March–April 2016).

**Mass balance analysis.** The mass balance analysis examines the different processes leading to methane gains and losses within the SML (Fig. 1). The gains include horizontal transport from the shore, vertical diffusion from the thermocline, river input and internal production (OMP). The losses are methane oxidation and surface emission and river outflow.

We used the following mass balance equation and solved either for oxic methane production, $P_{net}$ (= OMP), or lateral methane input, $F_L$[21]

$$\frac{\partial C}{\partial t} * \forall = (Q_R * C_R) + (Q_C * C_C) + (A_{th} * F_z) + (A_{sed} * F_L) + (P_{net} * \forall) \\ - (MOx * \forall + A_{tot} * F_S)$$

(1)

Here, $\frac{\partial C}{\partial t}$ describes the changing methane concentration over time [mol m⁻³ d⁻¹] (which under steady state condition is simplified to $\frac{\partial C}{\partial t} = 0$), $\forall$ is the volume of the surface mixed volume [m³]. ($Q_R \times C_R$) and ($Q_C \times C_C$) describes optional methane input and output by river in- and outflow where $Q_R$ ($Q_C$) is the flowrate [m³ d⁻¹] and $C_R$ ($C_C$) is the methane concentration of inflowing (outflowing) water [mol m⁻³]. The term ($A_{th} \times F_z$) describes the vertical methane input from below via interior turbulent diffusion: $F_z$ [mol m⁻² d⁻¹] ($z$ is the depth in a 1-m resolution) multiplied by the thermocline area $A_{th}$ [m²]. The term ($A_{sed} \times F_L$) describes lateral methane input from sediments with $A_{sed}$ being the surface area of the littoral sediment [m²] and $F_L$ being the sediment methane flux [mol m⁻² d⁻¹]. $P_{net}$ is the local methane production rate per unit SML volume [mol m⁻³ d⁻¹]. Methane loss terms include local oxidation rate ($MOx$; [mol m⁻³ d⁻¹]) and emission to the atmosphere ($A_{tot} \times F_S$; where $A_{tot}$ is the lakes' surface area [m²] and $F_S$ is the surface emission [mol m⁻² d⁻¹]). Note that $P_{net}$ symbolizes oxic methane production

which is abbreviated in the running text as OMP. The mass balance was parametrized accordingly (Supplementary Table 9).

**Monte Carlo simulation**. To assess uncertainties, Monte Carlo simulation was used (9999 iterations) when solving the mass balance. Using the rnorm-function of R[59,60], mass balance components were randomly picked within the normal distribution resulting from mean values ($\mu$) and their standard deviations $\sigma = \sqrt{\left(\left(\sum (x - \bar{x})^2\right)/(n-1)\right)}$ retrieved from field measurements. Here, the normal distribution has the density $f(x) = (1/\sqrt{2\pi}\sigma)e^{-((x-\mu)^2/(2\sigma^2))}$. Mass balance output is presented as mean $\pm 1\sigma$.

**Methane concentration**. In two experimental enclosures (1, 13) and the adjacent open-water in the South basin, methane concentration within the top 18 m of the water column was sampled in a 1-m resolution 4–5 times over 10 days in August 2014. Weekly water column profile sampling was also carried out between 10:00 and 18:00 local time, from March to July in 2016 at the open-water sites in the Northeast basin (69.5 m deep) and in the South basin (20.5 m deep). In July 2018, one additional profile measurement was taken in both basins. Furthermore, the central reservoir was sampled on three occasions in 2016 (on 3rd and 10th May when stratification was developing, and on 7th July when the water was fully stratified). Water was collected from different depths by a Limnos Water Sampler, and gently transferred to 50 ml serum bottles via a tubing. The bottles were fully flushed three times, filled and crimp-closed with PTFE-butyl septa (triplicates at the Northeast basin, duplicates elsewhere). Dissolved methane concentrations were measured in the lab by headspace displacement method and a GC/FID[61] (Shimadzu).

**Surface methane emission**. Methane surface emission ($F_S$) was captured by a 15 l-volume floating chamber. Trapped methane was quantified by withdrawing the gas from the chamber and measuring it by headspace analysis (GC/FID). Emission data were then used to derive gas transfer constant ($k_{600}$) as a function of wind speed at 10 m height ($U_{10}$) (Supplementary Note 2). For times when we did not have direct emission measurements, we used the $k_{600}$-relationship to estimate methane emissions based on wind speed. Parameters computed for flux estimations are summarized in Supplementary Table 4.

**Lateral methane input**. To estimate how much methane was introduced from littoral sediments into the SML during the stratified period, methane measurements were taken inside mesocosm enclosures (2 weeks after the water was exchanged with open lake water) and in the open water adjacent to the enclosures in the South basin (details in Supplementary Table 3). As the enclosures were cut off from lateral transport, by comparing the mass balance analysis results between inside and outside of the enclosures, we were able to derive the lateral methane input.

We neglected lateral methane input for the non-stratified season as sediment methanogenesis is highly temperature dependent[62,63] and was observed to be zero or 1–2 orders of magnitude smaller under winter conditions compared to summer/autumn condition[62,64,65].

**Vertical methane diffusion**. The stratified period (June–July) was characterized by a distinct methane peak in the thermocline. To estimate the transport of methane from the thermocline into the SML via (turbulent) diffusion, we applied the Fick's First law as follows

$$F_z = -K_z * \frac{\partial C}{\partial z} ; \left[ \text{mol m}^{-2}\text{d}^{-1} \right], \quad (2)$$

where $F_z$ is the average vertical methane diffusion, $z$ is depth [m], $\frac{\partial C}{\partial z}$ is the vertical methane gradient measured at 1-m depth resolution, and $K_z$ is the basin-scale diffusivity [m$^2$ s$^{-1}$] derived from temperature data based on the heat-budget method (Supplementary Note 5, Supplementary Fig. 3c). To obtain a conservative estimate of OMP in the SML, maximum $K_z$ values within the bottom 3 m of the SML were used to compute $F_z$. Temperature and diffusivity profiles measured inside the mesocosms were very similar to the open-water profiles allowing us to apply the same heat-budget estimates of open-water diffusivity values at depths >4 m to estimate the vertical flux in both open lake and mesocosm enclosures for the entire study period (Supplementary Fig. 8).

**Methane oxidation**. Methane oxidation ($MOx$) rates of up to 103 nmol l$^{-1}$ d$^{-1}$ have been observed in Lake Stechlin, when water was spiked with high methane concentrations[16]. However, $MOx$ rate in lake waters has been observed to differ by 1–2 orders of magnitude between winter and summer[45–47]. For a more conservative consideration ($MOx$ is a loss term in the mass balance) and to account for the seasonal difference and to simplify our mass balance analysis, we neglected $MOx$ for the non-stratified season, and we assumed

$MOx$ to be 30% of the internal production rate during the stratified season. We evaluated this assumption in a sensitivity analysis in the discussion section.

**River connection and ebullition**. Lake Stechlin is not connected to any river. Therefore, the corresponding mass balance terms ($Q_R \times C_R$) and ($Q_C \times C_C$) equal 0. No methane ebullition was observed during the whole study period. Earlier studies reported generally low methanogenesis activity in Lake Stechlin sediments[66–68], with the majority occurring below 20 cm sediment depth[69]. Tang et al.[16] demonstrated that ebullition did not contribute methane to SML waters for depths ≥20 m. This allowed us to ignore ebullition in our mass balance analysis for Lake Stechlin (22.7 m mean depth).

**Environmental parameters**. Water depths were measured by a portable sounder gauge (Cole-Parmer). Temperature, dissolved oxygen and chlorophyll fluorescence was measured using a YSI probe (Model 6600V2). Wind speed data ($U_{10}$ recorded at 10 m height) were provided in 30–60 min resolution by the Neuglobsow weather station (Federal Environmental Agency) adjacent to the lake.

**Oxic methane contribution**. We examined the importance of oxic methane production relative to anoxic sources (lateral input, vertical diffusion) by computing the OMC

$$OMC = (P_{net} * \forall) * 100/((P_{net} * \forall) + (A_{sed} * F_L) + (A_{th} * F_z)); [\%]. \quad (3)$$

We then compared our results with the literature data[21,32] (Supplementary Note 3) to examine $OMC$ as a function of lake morphology. To expand our analysis to larger lakes, we estimated $OMC$ for additional lakes based on the data in DelSontro et al.[39] (Supplementary Note 3, Supplementary Table 6).

**Data format**. This study contains multiple field samplings done in the course of 2014, 2016, and 2018. Mean ± 1 standard deviations presented throughout the manuscript indicate temporal variation and were calculated separately for the stratified/non-stratified season for each basin or combined for the experimental enclosures or the central reservoir. $R^2$ values presented throughout the paper are based on LM models.

**Reporting summary**. Further information on experimental design is available in the Nature Research Reporting Summary linked to this paper.

## Data availability
Data are made available in graphical or tabular form throughout the paper and Supplementary Information. The source data underlaying Figs. 2–4 and Supplementary Figs. 1, 3, 4, 5, 6, and 8 are provided as a Source Data file.

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

## Acknowledgements

We thank Anke Penzlin and Marcus Wallasch (Umweltbundesamt, Germany) kindly provided weather data, Peter Casper (Leibniz Institute, Germany) for giving access to a GC/FID unit and the Lake Lab team (Leibniz Institute, Germany) for giving access to the Lake Lab enclosures and automated profiler data. Further thanks to Matthew Bogard (University of Washington Seattle, USA) for providing morphology parameters of Lake Cromwell and Tonya DelSontro (University of Geneva, Switzerland) for providing lake data for *OMC* estimations. Funding was provided by the Swiss National Science Foundation for D.F.M and D.D. (grant 200021_169899), by the German Research Foundation for G.K. (KI-853/7-1, KI-853/11-1, KI-853/11-2). D.I., M.B. and H.P.G. were funded by the German Research Foundation (DFG; GR1540/21-1+2, 23-1, 28-1, BI1987/2-1), the German Federal Ministry of Education and Research (BMBF 01LC1501G) and the European Commission/ Horizon program (H2020 project ERA-PLANET).

## Author contributions

M.G., H.-P.G., K.W.T., D.F.M., D.D. and G.K. contributed to the design of the study. M.G., H.-P.G., D.I. and M.B. collected the data; M.G. analyzed the data with input from D.F.M., D.D. and G.K.; M.G. and K.W.T. wrote the paper with input from all co-authors.

## Competing interests

The authors declare no competing interests.
