## [Peer Review File · Nature Communications]

Reviewers' comments:

Reviewer #1 (Remarks to the Author):

Review of "Contribution of oxic methane production to surface methane emission in lakes – local and global importance"
by Günthel et al., submitted to Nature Communication

The authors present estimates of oxic methane production in lake and its contribution to surface to atmosphere flux, using a mass balance analysis based on in-situ measurements. The measurements have been performed all year round (allowing seasonal variation study) and in two basins of one lake. Complementary analysis has been performed in an artificial laboratory lake. In their study the authors estimate that oxic methane production is an important contribution to surface-water emissions (more than 50% for this lake). They also determine a predictive function of oxic methane production contribution depending on littoral sediment area and surface mixed layer volume.

GENERAL COMMENT

The manuscript is well written. The methodology and different steps are well explained. Processes involved in methane production in lakes and reservoirs are still highly debated in the community. Understanding and estimating the different processes and transport pathways need further work and proofs to unify the community and produce a better knowledge on methane production and transport in these freshwater systems. This study contributes significantly to the debate. However, the global perspective needs to be further discussed by highlighting the uncertainty in estimating methane emissions from lakes and the reasons why. How such a study could help reducing this uncertainty and helping estimating methane emissions at the global scale? Some suggestions are provided in the specific comments but the authors may have other ones. In the text, the measurements used in this study are not well specified – though it is in the different Tables and Figures. Maybe one or few sentences at the beginning may help clarify whether one or several years of data are used in order to complete the missing details on the instruments and data collected (number of flux chamber for example), just one vertical profile (one location or several)... Going from local to global estimates is a tricky step and needs to acknowledge some uncertainty during to up-scaling process. Finally, I would recommend publication in Nature Communication after minor revisions addressing the general and specific comments highlighted in this review.

SPECIFIC COMMENT

Abstract:

l. 21: "freshwater", what does it refer to exactly here? Lakes and rivers, and ponds, and reservoirs.. ??

Introduction

l. 33: "second most carbon based GHG". This is a bit awkward.

l. 36: "frequently observed methane accumulation". How frequent is this? Please include references to this statement. Could you add any references to "first" studies presenting this paradox?

l. 44: The estimate of freshwater emissions is highly uncertain – large uncertainty (Saunio et al., 2018, report a minimal uncertainty range). Also the estimates suggested by the literature, when added to the other known emissions (both anthropogenic and natural) lead to total methane emissions inconsistent with the atmospheric burden – (and what is known from the methane sinks). This large uncertainty of freshwater emissions and the different reasons (accurate surface area, types of freshwater system, methane flux density used to upscale the local to global estimates) need to be acknowledge here. Indeed a better understanding of the production and transport pathways of methane in lakes and reservoirs is helpful to better constrain the water-to-surface methane emissions.

l. 51: Why exactly IPCC reports do not properly include OMP? Note that the first draft of the next IPCC

report from group I is available for comments. This missing point may be highlighted and corrected for the next report.

I. 68-69: here the authors claim that long term monitoring has been performed in this lake, and that they have detailed data since 2010. However they present results only for year 2016 and 2018 (Fig 3). It is not clear if the results use only the 2016 and 2018 measurements or if any measurements from the other years are used. Sentence I. 68-70 suggests this but this is not clear from the following text. Clarify how the data have been used and if mass-balance is possible for data from 2010, allowing inter annual variability assessment. If data before 2016 are not used, explain why.

RESULTS

I. 104: here and elsewhere (Text, Table, fig, and all supplementary material): April of which year?? 2016? 2018? Same month for all years? Please specify to which year and data your results correspond.

I. 110: define the study period at the beginning to clarify the study.

I. 144: water to surface emissions have been measured using a flux chamber. Does it mean that only one flux chamber was used in a single point of the basin? How the spatial variability over the basin may influence the results? Same question for the concentration profiles (Fig 2). How the location for these measurements has been chosen? One bias of a box model is that spatial variability is not fully taken into account. How spatial variability would impact on the results of this study?

I. 149. Here "other sampling sites" are mentioned. Please clarify the experimental set up, a bit in the text and in more detail in the supplementary.

I. 151: here and below, on which basis is the standard deviation calculated? Does it represent the temporal variation inside each stratified/non-stratified period?

I.153: Reason for the difference between the two basins?

I. 169: Reason(s) for the difference between open water and open-enclosures?

I. 173: Refer Eq. 1 to Method Section for clarity. OMP does not clearly appear in Eq 1, which makes things harder to follow for non-specialist. Here specify that this value was determined for early August.

I. 182: Does it mean that the value $Fl = 76 \pm 12 \text{ nmol l}^{-1}\text{d}^{-1}$, found for August is used for the whole stratified period? Or that the same calculation is done for each weekly measurement?

I. 188 to 193: The Monte Carlo simulations result in a large standard deviation. Is the SD reported 1sigma? 2sigma? Which confidence percentiles? The SDs are as large as or larger than the derived value. Could the authors comment on these large uncertainties? Similarity to previous studies? Relevance of results?

Table1: give the definition of the SD

DISCUSSION.

I. 223: Rephrase "3) the substantial contribution ... methane emission (64%..)

I. 226 to 230. The model discussed here has not been mentioned previously nor its results. More details should be given to strengthen this statement.

I. 240: Please specify the previously reported range. Does the SD measured in this study fit the range?

I. 242-243: what is the point here?

I. 244: could the authors comment on the differences between the two basins?

I. 248: refer to Table S5.

I. 254: Specify the previously reported range.

I.357-358 and I. 379: Recall 1) the large uncertainty and lack of knowledge on the processes and transport pathways 2) the uncertainty in individual lake measurements and issues related to upscaling (definition of lakes, ponds, reservoirs, uncertainty on surface areas...). A better understanding of the processes and transport pathways may allow a better modelling approaches to derive other estimates of methane emissions from lakes, and avoid issues of upscaling, as currently done. This might help for a better assessment of methane emissions from lakes and reservoirs, though the values derived here for a single lake, are highly uncertain. More comment and discussion on the global perspective and methane budget is needed to highlight the added value of defining novel predictive models.

Reviewer #2 (Remarks to the Author):

1. This paper reinforces the importance of oxic methane in lake methane emission, and analyzes the contribution of oxic methane production to surface methane emission in two portions of a temperate lake, and with a hope to project its global contributions with additional few lakes in the literature. The main contribution of this manuscript would be on its global context.
2. This paper is well written.
3. Lake Stechlin does not have major inflows or outflows. However, most freshwater lakes globally have flows. The representativeness of this lake is low in the global context.
4. Two basins? How there are many ways of cutting portions. The separation might not affect the mass balancing model but would change the global prediction relationship. In general, I do not agree using the two portions of the same lake as two individual lakes in the subsequent analysis.
5. Line 144. Surface methane emissions other than the Northeast portion were derived from wind speed, based on a linear relationship, but not so strong ($r^2 = 0.44$). How would this impact on the balancing analysis?
6. Fig. 3. Methane concentration jumps quickly in early stratified state and drops back quickly in the rest of the state. However, the methane measurements were made in the early state not throughout the entire stratified state. This could lead to substantial over-estimation on OMP and bias in the analysis.
7. Lines 188-192. The OMP rates at 72 ± 74 and 88 ± 75 are of very high uncertainty. How much is the uncertainty to the 64% and 50%?
8. Line 277. How many lakes were evaluated for the range between 4 and 30 $\text{nmol l}^{-1} \text{d}^{-1}$? What are these lakes' characteristics?
9. Line 278. "MOx to be equivalent to a constant fraction (30 %) of the internal production". This arbitrary treatment would need a justification.
10. Again, there are too many simplified treatments lack of reasonable justifications.
11. In the global prediction, it is inappropriate to use the two portions of the lake as two lakes in the predictive model. Even the same lake has very different methane concentration, indicating lake size is not sufficient. There are many other factors to be considered such as flow conditions, wind fields, carbon richness in lakebed, lake shape and depth, and so on, in addition to size.
12. The global extrapolation is projected from only a few lakes found in the literature. However, their representativeness to global lakes is not discussed. Large variations many found among these selected lakes. We can imagine the huge discrepancy of lakes across the Earth, given the wide variety of landscape, lake history, flow connections, etc. This over-simplified extrapolation is questionable and unconvincing.
13. Line 66. After this study, it still does not seem to be clear on "what drives the underlying processes and what causes different contribution patterns in different lakes." Otherwise, the global predictive model would be more sophisticated.

Point-by-point response to the reviewers' comments on the manuscript: "Contribution of oxic methane production to surface methane emission in lakes – local and global importance" by Günthel and colleagues

NB: After careful consideration of the reviewers' comments, the title has been changed to "Contribution of oxic methane production to surface methane emission in lakes – local and global implications".

1) Reviewer #1 (Remarks to the Author):

The authors present estimates of oxic methane production in lake and its contribution to surface to atmosphere flux, using a mass balance analysis based on in-situ measurements. The measurements have been performed all year round (allowing seasonal variation study) and in two basins of one lake. Complementary analysis has been performed in an artificial laboratory lake.

In their study the authors estimate that oxic methane production is an important contribution to surface-water emissions (more than 50% for this lake). They also determine a predictive function of oxic methane production contribution depending on littoral sediment area and surface mixed layer volume.

GENERAL COMMENT

1. The manuscript is well written. The methodology and different steps are well explained. Processes involved in methane production in lakes and reservoirs are still highly debated in the community. Understanding and estimating the different processes and transport pathways need further work and proofs to unify the community and produce a better knowledge on methane production and transport in these freshwater systems. This study contributes significantly to the debate.

Response: We appreciate the compliments.

2. However, the global perspective needs to be further discussed by highlighting the uncertainty in estimating methane emissions from lakes and the reasons why. How such a study could help reducing this uncertainty and helping estimating methane emissions at the global scale? Some suggestions are provided in the specific comments but the authors may have other ones.

Response: The revised manuscript highlights the uncertainty in global assessments of methane emissions, the corresponding causes, and how this study helps reduce this uncertainty. Please see introduction (l. 48-66) and discussion (l. 386-397) in the revised manuscript.

l. 48-66: "Globally, it is estimated that freshwaters account for (SD \pm minimum error range) 122 \pm 60 Tg yr⁻¹ methane to the atmosphere (ca. 20 % of the total emission)²². However, this emission value is not well constrained as indicated by the 100 % minimum uncertainty range²², and contributes to

bottom-up and top-down methane budgets not agreeing with each other^{22,23}. The large uncertainty of freshwater emission during upscaling is commonly attributed to i) highly variable methane density fluxes within and across systems²⁴⁻²⁷, ii) scarce long-term data, which do not cover high ecosystem variability^{22,28}, or iii) uncertainties in global freshwater areas²⁹⁻³¹. Oxidic methane production has so far not been considered in global assessments including methane budgets^{22,23} and IPCC reports^{1,2} despite its potential to cause highly variable methane density fluxes in freshwater systems^{15,21,32}. For more accurate modelling of freshwater emission and corresponding contribution to the global methane budget, a better understanding of internal methane production, consumption and distribution pathways is needed.

While methanogenic Archaea are largely responsible for anoxic methane production^{6,33}, primary production has been associated with the oxidic methane source^{15,17,32,34}. Therefore, the oxidic and anoxic sources will react differently to environmental factors. Global methane budget assessments and future climate change predictions will benefit from proper distinction of oxidic versus anoxic methane sources and identifying their individual contribution to the system-wide emission.”

I. 385-396: “In addition to known knowledge gaps in the global methane dynamics^{22,23}, OMP has not been considered as source of uncertainty in global assessments^{1,2,22,23}. Because both oxidic and anoxic methane sources in lakes can be modulated by multiple factors and processes (Supplementary Fig. 7), some of which are still poorly understood, it would be premature to construct a mechanistic model to fully describe methane dynamics in lakes. Instead, we developed empirical models as useful tools to predict the contribution of OMP to the system-wide emission (OMC) in stratified meso-to-oligotrophic lakes in the temperate region based on a set of simple lake morphological parameters (Fig. 4, Supplementary Fig. 6).

The first model using littoral sediment area (A_{sed}) and SML volume (V) as proxy explains nearly the entire variance in the dataset ($R^2 = 0.95$, $p \ll 0.01$) making it a powerful predictive model to estimate OMC from A_{sed} and V .”

3. In the text, the measurements used in this study are not well specified – though it is in the different Tables and Figures. Maybe one or few sentences at the beginning may help clarify whether one or several years of data are used in order to complete the missing details on the instruments and data collected (number of flux chamber for example), just one vertical profile (one location or several)...

Response: We apologize for not stating this more clearly. We have added a detailed description of methane measurements in Supplementary Table 3, which is referenced on several occasions. In addition, we have now specified in the introductory section which data sets (years) were used (l. 79-88).

Supplementary Table 3 | Sampling schedule throughout 2014 – 2018. Detailed descriptions on how parameters were recorded can be found in the method section. Measurements in the experimental enclosures were taken 2 weeks after the water had been exchanged with lake water; in contrast, water in the central reservoir has never been exchanged since its installation in 2012. Sampling locations are further described in Supplementary Figure 2.

Year	Month	Location	Purpose	n ^a	WC Profiles ^b	Surface emission ^c	Environmental parameters ^d
2014	Aug	exp. enclosure 1	quantify F_L	4 (SS)	yes	modelled	yes
2014	Aug	exp. enclosure 13	quantify F_L	5 (SS)	yes	modelled	yes
2014	Aug	South basin	quantify F_L	4 (SS)	yes	modelled	yes
2016	Mar-Jul	North basin	seasonal OMP,	6 (SS)	yes	measured (5/6)	yes
			basin variation	13 (NS)	yes	measured (13/13)	yes
2016	Mar-Jul	South basin	seasonal OMP,	6 (SS)	yes	modelled	yes
			basin variation	10 (NS)	yes	modelled	yes
2016	May, Jul	central reservoir	seasonal OMP,	1 (SS)	yes	modelled	yes
			isolated water	2 (NS)	yes	modelled	yes
2018	Jul	North basin	seasonal OMP,	1 (SS)	yes	measured (1/1)	yes
			basin variation				
2018	Jul	South basin	seasonal OMP,	1 (SS)	yes	measured (1/1)	yes
			basin variation				

(a) n represents the repetition of methane measurements (each taken on a different day during day time) including water column profile (water samples transferred into glass bottles, crimp-closed, He head space replacement, GC/FID analysis) and surface emission (floating chamber measurements), recordings were taken during the stratified season (SS) or non-stratified/intermediate season (NS); (b) WC profiles indicate water column methane profiles that were taken from the surface down to below the thermocline (ca. 5-7 m depth) in 1 to 2 m increments; (c) surface emission was measured (see methods) using a floating chamber or estimated from a wind based model developed from our own floating chamber measurements and compared to models in the literature (details in Supplementary Note 2); (d) environmental parameters include wind data that were recorded in 10 m above lake surface by the Neuglobsow weather station next to Lake Stechlin and were provided by the Umweltbundesamt, water temperature was recorded by automated YSI probes permanently mounted on the lake lab facility in the South basin (profiling the upper 20 m of the water column continuously in 60 min intervals); F_L is the lateral methane input and OMP is oxic methane production; in 2016 Lake Stechlin stratified ca. mid-May (see Supplementary Fig. 3a,b); samplings in 2014 and 2018 were done during stratification.

I. 79-88: “Throughout the years 2014, 2016 and 2018 we collected a detailed data set of dissolved methane concentration, surface methane emission and environmental parameters (temperature, dissolved oxygen, algal pigments, wind speed) including different sampling sites: Northeast and South basins and inside enclosures (see Supplementary Table 3 for data overview). This dataset allowed us to conduct a detailed methane mass balance analysis for the surface mixed layer, accounting for the different sources and sinks (Fig. 1), including lateral methane input (2014 dataset) and OMP (all datasets), under different seasonal conditions (2016 dataset, 1 repetition in 2018). Datasets collected before 2014 in Lake Stechlin do not include all information necessary for mass balance analysis^{15,16}; however, we compare our mass balance results to earlier findings.”

4. Going from local to global estimates is a tricky step and needs to acknowledge some uncertainty during to up-scaling process.

Response: We have now acknowledged the uncertainty in upscaling in the manuscript. Supplementary Fig. 7 highlights various factors affecting methane cycling in lakes. General challenges in upscaling to the global methane budget are now explained in the introduction

including the contribution by OMP (l. 48-66, 385-396). Furthermore, lake parameters that potentially influence the empirical model(s) are discussed in more detail (l. 401-406, l. 410-415). We clarify that our study did not aim to build a mechanistic model; rather, we aimed to develop statistical models as useful tools to predict the contribution of OMP to surface emission in lakes based on simple lake morphology characteristics (l. 391-394).

In response to both reviewers' comments we have changed the title to 'Contribution of oxic methane production to surface methane emission in lakes – local and global **implications**'.

l. 48-66: *quotation in response to comment 2*

l. 385-396: *quotation in response to comment 2*

l. 401-406: "The system-wide contribution of the anoxic methane sources is mainly controlled by littoral sediment flux and the corresponding littoral sediment area. Trophic state^{52,53} and temperature^{12,54} are important drivers of the methane flux from sediments. Higher sediment methane fluxes in eutrophic systems and in warmer climate zones compared to our dataset of stratified meso-to-oligotrophic lakes in the temperate region could shift the curve of the empirical models to the right (Fig. 4, Supplementary Fig. 6)."

l. 410-415: "The OMC prediction, therefore, may vary mainly for small lakes which have been reported to cause less methane emission on a global scale compared to large lakes²⁸ (<0.01 versus >1 km²). It shall be noted that the model predictions based on A_{sed} and \forall will be more reliable than based exclusively on lake surface area due to sediment steepness/aspect ratio and total depth modulating the littoral sediment area at constant lake surface area."

l. 391-394: "...we developed empirical models as useful tools to predict the contribution of OMP to the system-wide emission (OMC) in stratified meso-to-oligotrophic lakes in the temperate region based on a set of simple lake morphological parameters..."

Supplementary Fig. 7 is depicted on the next page.

Supplementary Figure 7 | Examples of factors affecting the contribution of oxic and anoxic methane sources to the system-wide surface emission. Factors are categorized into morphology, sediment characteristics, nutrient conditions/ecology, meteorology and lake physics. A_{sed} symbolizes littoral sediment area, SML is surface mixed layer, V refers to the volume of the surface mixed layer and Q is flow rate.

5. Finally, I would recommend publication in Nature Communication after minor revisions addressing the general and specific comments highlighted in this review.

Response: We are grateful for the constructive comments. We have introduced changes in accordance to all the comments, giving a more comprehensive presentation and understanding of our study. These changes include more detailed introduction discussion sections, and the presentation of the models has been moved to the result section to improve the flow and to better highlight this key part of the study. Please note, the order of references and supplementary materials has also changed in the revised manuscript.

SPECIFIC COMMENT

Abstract:

6. l. 21: “freshwater”, what does it refer to exactly here? Lakes and rivers, and ponds, and reservoirs.. ??

Response: We have changed the phrase from ‘freshwater’ to ‘sea and lake waters, as well as wetlands’ as these are the primary aquatic systems in which oxic methane production has been observed and studied so far (l. 21-22).

Introduction:

7. l. 33: “second most carbon based GHG”. This is a bit awkward.

Response: We have changed the phrase to ‘second most important carbon-based GHG’ (l. 33-34).

8. l. 36: “frequently observed methane accumulation”. How frequent is this? Please include references to this statement. Could you add any references to “first” studies presenting this paradox?

Response: We reference the review article by Tang et al. (2016), which lists over 20 examples of reports of oversaturated methane concentrations in oxic sea and lake waters, including the earliest report (to our knowledge) by Scranton & Brewer (1977) (l. 36-38).

l. 36-38: “Over the past several decades⁷ there have been multiple reports of paradoxical methane oversaturation in oxic sea and lake waters (Tang et al.⁸ and references herein).”

Tang et al. 2016, *Environ. Sci. Technol. Lett.* **3**, 227-233

Scranton & Brewer 1977, *Deep-Sea Res.* **24**, 127-138

9. l. 44: The estimate of freshwater emissions is highly uncertain – large uncertainty (Saunois et al., 2018, report a minimal uncertainty range). Also the estimates suggested by the literature, when added to the other known emissions (both anthropogenic and natural) lead to total methane emissions inconsistent with the atmospheric burden – (and what is known from the methane sinks). This large uncertainty of freshwater emissions and the different reasons (accurate surface area, types of freshwater system, methane flux density used to upscale the local to global estimates) need to be acknowledge here. Indeed a better understanding of the production and transport pathways of methane in lakes and reservoirs is helpful to better constrain the water-to-surface methane emissions.

Response: In response to this and comment 4, we have revised the introduction now acknowledging the uncertainties in the upscaling exercises and explained how our study contributes to the improvement of global assessments (l. 48-66).

I. 48-66: quotation in response to comment 2

10. I. 51: Why exactly IPCC reports do not properly include OMP? Note that the first draft of the next IPCC report from group I is available for comments. This missing point may be highlighted and corrected for the next report.

Response: This is correct that oxic methane production so far has not been considered in IPCC reports. For examples:

Climate Change- the IPCC scientific assessment (IPCC 1990) p.18 states: **“Methane is a chemically and radiatively active trace gas that is produced from a wide variety of anaerobic (i.e., oxygen deficient) processes...”**

and the most recent *Climate Change 2013 summary for policymakers- the physical science basis (IPCC 2013)* p.141 states: **“Where oxygen is limited, as in waterlogged soils, some microbes also produce methane.”**

In the IPCC special report *Climate Change and Land* (released August 2019), section 2.3.2 *Methane* discusses the atmospheric methane trends, methane sources in different biomes (e.g. soils, wetland, peatland) and land use effects; the special report does not mention oxic vs. anoxic methane production or attempt to make such a distinction. The sixth assessment *AR6 Climate Change 2021: The Physical Science Basis* is still being compiled and is expected to be finalised in 2021. We are aware that the first draft of WG1 contribution to AR6 was opened for expert review (29 April to 23 June 2019); however, our understanding is that expert review should be based on published papers, and therefore we try to get our work published in a timely manner.

The amount of research and papers on OMP pales in comparison with conventional anoxic methane production, and our understanding of this new phenomenon is still limited. Because IPCC reports often draw from the more ‘established’ science and the reports serve the important purpose of guiding long-term national and international climate policies, it is perhaps understandable that the reports focus only on the conventional anaerobic/anoxic methane sources.

It is our hope that this manuscript will add to the growing body of knowledge about OMP and stimulate more research and discussion on the topic, and that as our understanding of OMP continues to increase, it will be included in future IPCC reports.

Climate Change-The IPCC Scientific Assessment. Houghton, J.T., Jenkins, G.J. and Ephraums, J.J. (eds.) Cambridge University Press, Cambridge (1990).

IPCC, Summary for Policymakers. In: *Climate Change 2013: The Physical Science Basis. Contribution of Working Group I to the Fifth Assessment Report of the Intergovernmental Panel on Climate Change (IPCC)* [Stocker, T.F., D. Qin, G.-K. Plattner, M. Tignor, S.K. Allen, J. Boschung, A. Nauels, Y. Xia, V. Bex and P.M. Midgley (eds.)]. Cambridge University Press, Cambridge, United Kingdom and New York, NY, USA, 1535 pp.

IPCC, *Climate Change and Land* (07 August 2019): An IPCC Special Report on climate change, desertification, land degradation, sustainable land management, food security, and greenhouse gas fluxes in terrestrial ecosystems.

11. I. 68-69: here the authors claim that long term monitoring has been performed in this lake, and that they have detailed data since 2010. However they present results only for year 2016 and 2018 (Fig 3). It is not clear if the results use only the 2016 and 2018 measurements or if any measurements from the other years are used. Sentence I. 68-70 suggests this but this is not clear

from the following text. Clarify how the data have been used and if mass-balance is possible for data from 2010, allowing inter annual variability assessment. If data before 2016 are not used, explain why.

Response: We apologise for not making this clearer. While we have basic environmental data for Lake Stechlin from an earlier time, some of the parameters needed for the methane mass balance analysis are not available before 2014. For the purpose of this study, we use data taken in Aug 2014 (South basin, enclosures), in Mar-Jul 2016 (South and Northeast basin, central reservoir) and Jul 2018 (South and Northeast basin). We have clarified the time period in the text (l. 79-88) and inserted a summary in Supplementary Table 3.

Both **Supplementary Table 3** and quotations of **l. 79-88** are stated in response to comment 3.

RESULTS

12. l. 104: here and elsewhere (Text, Table, fig, and all supplementary material): April of which year?? 2016? 2018? Same month for all years? Please specify to which year and data your results correspond.

Response: We apologise for omitting time references. April refers to the 2016 dataset. We have included this information throughout the revised manuscript. Please see Supplementary Table 3, which gives a detailed overview about the time of measurements.

Supplementary Table 3 is included into the response to comment 3.

13. l. 110: define the study period at the beginning to clarify the study.

Response: As suggested, we have defined the study period in the introduction and in more detail in Supplementary Table 3 (l. 79-88).

See response to comment 3 for details.

14. l. 144: water to surface emissions have been measured using a flux chamber. Does it mean that only one flux chamber was used in a single point of the basin? How the spatial variability over the basin may influence the results? Same question for the concentration profiles (Fig 2). How the location for these measurements has been chosen? One bias of a box model is that spatial variability is not fully taken into account. How spatial variability would impact on the results of this study?

Response (*'Does it mean that only one flux chamber was used in a single point of the basin?'*): The mass balance analysis was conducted for two basins using different methods: In the Northeast basin surface methane emission was measured using a floating chamber deployed at the deepest point. Emission from the South basin was calculated using a wind-based model; the calculated emission values were checked against multiple published models that are mainly based on direct turbulence measurements and that account for variability caused by wind speed, heat flux and basin size (see sensitivity analysis throughout the discussion; l. 305-324).

l. 305-324: "The average surface methane emission during the stratified period was $0.47 \text{ mmol m}^{-2} \text{ d}^{-1}$ ($\pm 57 \%$ SD) in the Northeast basin and $0.71 \text{ mmol m}^{-2} \text{ d}^{-1}$ ($\pm 34 \%$ SD) in the South basin (taken mainly during calm weather). The larger value in the South basin can be attributed to higher influence from littoral methane sources. However, these emission values are comparable with the global estimate of $0.62 \text{ mmol m}^{-2} \text{ d}^{-1}$ for the region $25\text{-}54^\circ$ latitude⁴² and within the range reported earlier for Lake Stechlin⁴³ (exceeding $4 \text{ mmol m}^{-2} \text{ d}^{-1}$ at strong wind; $2.6 \text{ mmol m}^{-2} \text{ d}^{-1} \pm 42 \%$ SD). Highly variable surface emission has been reported earlier, for some systems standard deviations

exceed 100 % of mean emission values during summer^{24,26}. In case of the South basin we estimated the emission from wind speed data and the corresponding results are dependent on the gas transfer constant (k_{600}) value used. Our k_{600} -wind speed relationship ($k_{600}[\text{cm h}^{-1}] = 1.98 \cdot U_{10}[\text{m s}^{-1}] + 0.98$) was very similar to an earlier report (e.g. Lake Hallwil: $k_{600}[\text{cm h}^{-1}] = 2.0 \cdot U_{10}[\text{m s}^{-1}]$; Donis et al.²¹). Applying six alternative emission models (based on wind or combined wind and lake size) presented by Vachon and Prairie³⁷, MacIntyre et al.³⁸ and Donis et al.²¹ to this dataset resulted in an average emission rate between 0.55 and 1.03 $\text{mmol m}^{-2} \text{d}^{-1}$. Applying these alternative emission rates to the mass balance analysis gave an OMP rate between 41 and 185 $\text{nmol l}^{-1} \text{d}^{-1}$, which still translated to a substantial oxic methane contribution (32 – 68 %) to the surface methane emission (details in Supplementary Table 7). In other words, regardless of the method or model used to estimate surface methane emission, it remains that OMP was an important contributor to surface emission.”

Response (*‘How the spatial variability over the basin may influence the results? Same question for the concentration profiles (Fig 2)’*): The main sources of spatial variabilities in surface mixed layer methane concentration and surface emission are vertical (mainly ebullitive) and lateral (diffusive) transport from anoxic sources. From earlier studies we know that ebullition mainly contributes to the surface mixed layer at depths below 20 m (Tang et al. 2014). Lake Stechlin, with a mean depth of 22.7 m, has low methanogenesis activity mainly occurring below 20 cm in the sediments (Casper 1996, Casper et al. 2003/2005, Conrad et al. 2007), making ebullition-based variability less important. This is consistent with methane sampling done at various locations in Lake Stechlin and with transect measurements presented in earlier studies (Tang et al. 2014, Bizic Ionescu et al. 2019) indicating no strong spatial variability in Lake Stechlin.

Recently, DelSontro and colleagues (2018) analysed the horizontal transport of methane from shore to the basin’s centre for a range of lakes. Their results show that the horizontal transport in the surface mixed layer follows a predictive function and depends on the shore-distance: Littoral sediment methane strongly contributes to the in situ profile and emission close to shore, and the contribution decreases with increasing distance; at distances above ca. 2 km there is no effective contribution anymore. In our study we used mesocosm enclosures to determine the lateral methane source strength, providing average littoral sediment production rates.

For locations closer to the shore side, more methane from the shore would have caused higher water column methane, which would have also elevated the surface emission. The advantage of our mass balance approach is that we can account for vertical transport (measured in situ profile, Fick’s Law), lateral transport (measured by mesocosm enclosures) and surface emission (measured by flux chambers or calculated from models) at the sampling location, and that our approach, therefore, also accounts for the shore-centre-distance relationship.

To further account for any unknown anoxic sources, we conducted a sensitivity analysis, which shows that the oxic methane source was still a substantial source even if the surface emission would have been overestimated by ca. 30 %. In other words, even if we allowed for variability and error in our mass balance analysis, the overall conclusion remained the same, e.g. OMP was an importance source of methane surface emission.

Response (*‘How the location for these measurements has been chosen?’*): Our study focusses on the mid-water column. To get the most unbiased result (e.g. bias from spatial variability in littoral sediment methanogenesis), the most remote place should be sampled, which is the basin’s centre and often deepest point. In the North-East basin we conducted the sampling at the deepest point (ca. 69.5 m). In the South basin we sampled at a 20 m deep site near the mesocosm enclosures to

get the best comparison between enclosure and open lake. Computed oxic methane production rates in the Northeast and South basin are very comparable supporting the sampling locations to be representative of both basins.

Response (*One bias of a box model is that spatial variability is not fully taken into account. How spatial variability would impact on the results of this study?*): As explained above, we conducted a sensitivity analysis, which shows that the oxic methane source remained a substantial source even if the surface emission would have been overestimated by ca. 30 %.

Tang et al. 2014, *Limnol. Oceanogr.* **59**, 275-284

Casper et al. 2003, *Arch Hydrobiol. Spec. Issues Adv. Limnol.* **58**, 53-71

Casper et a. 2005, *Verh. Int. Ver. Limnol.* **29**, 564-566

Conrad et al. 2007, *Limnol. Oceanogr.* **52**, 1393-1406

Casper 1996, *Arch. Hydrobiol. Spec. Issues Advanc. Limnol.* **48**, 253-259

DelSontro et al. 2018, *Ecosystems* **21**, 1073-1087

Bizic-Ionescu et al. 2018, Springer International Publishing AG, A. J. M. Stams, D. Z. Sousa (eds.), Biogenesis of Hydrocarbons, Handbook of Hydrocarbon and Lipid Microbiology

15. I. 149. Here “other sampling sites” are mentioned. Please clarify the experimental set up, a bit in the text and in more detail in the supplementary.

Response: We have clarified that flux chamber measurements were done in the Northeast basin and wind-based models were used to estimate emissions in the South basin (enclosures and open lake). We have added the information how we used models from the literature to validate the emission values (I. 167-171). The new Supplementary Table 3 explains in more detail which data sets contain measured and modelled values.

I.167-171: “This linear function was then used to estimate surface emissions in the South basin (enclosures and open lake) based on wind speed (Supplementary Note 2, Supplementary Table 4). Other published models^{21,37,38} in the literature (mainly based on direct turbulence measurements)^{37,38} were used to validate these emission values (see sensitivity analysis in Discussion).”

16. I. 151: here and below, on which basis is the standard deviation calculated? Does it represent the temporal variation inside each stratified/non-stratified period?

Response: Yes, the mean and standard deviations given throughout the manuscript indicate temporal variations of replicate measurements done either during the stratified or non-stratified period, separately for each basin/combined for the experimental enclosures/the central reservoir. We have added this information to the method section under ‘data format’ (I. 568-571).

I.568-571: “This study contains multiple field samplings done in the course of 2014, 2016 and 2018. Mean \pm 1*standard deviations presented throughout the manuscript indicate temporal variation and were calculated separately for the stratified/non-stratified season for each basin or combined for the experimental enclosures or the central reservoir.”

17. I.153: Reason for the difference between the two basins?

Response: Higher methane surface emissions were recorded in the South basin compared to the Northeast basin. This result is based on different proximity of the sampling sites to the littoral zone and different basin morphology. As we explained in our response to point 14, the influence of anoxic littoral methane sources increased closer to the shore. Consequently, the water column in the South basin contained more anoxic methane (Fig. 2) leading to higher surface emission of methane (Supplementary Table 4). Further, the South basin is scaled to 1.12 km² surface area with ca. 0.31 km² littoral sediment area while the Northeast basin is scaled to 2.01 km² with ca. 0.28 km² littoral sediment area. With the same amount of littoral sediment area but half the size, more anoxic methane would reach the South basin's mid-water column and cause higher surface emissions.

18. I. 169: Reason(s) for the difference between open water and open-enclosures?

Response: Comparing the vertical methane transport from below the thermocline into the surface mixed layer gave higher values in the open water versus the experimental enclosures. The experimental enclosures are cut-off from lateral methane input. The missing lateral source leads to less methane in the surface mixed layer compared to the open water (see Fig. 2). Additionally, the weaker methane gradient across the thermocline in the experimental enclosures led to less vertical methane diffusion.

19. I. 173: Refer Eq. 1 to Method Section for clarity. OMP does not clearly appear in Eq 1, which makes things harder to follow for non-specialist. Here specify that this value was determined for early August.

Response: As suggested by the reviewer, we have made a reference to the method section (I. 196-197, I. 204-206) and P_{net} has been defined as OMP (I. 465, 477-478). Information has been added to the introduction for the times when the corresponding measurements were made in the experimental enclosures and the open lake (I. 79-88, Supplementary Table 3).

I. 196-197: "...therefore OMP in the SML was estimated from Eq. 1 (see method section)"

I. 204-206: "Oxic methane production at high temporal resolution (approximately weekly) in the two open-water sites was estimated from Eq. 1 (see method section)..."

I. 465: "... P_{net} (= OMP)..."

I. 477-478: „Note that P_{net} symbolizes oxic methane production which is abbreviated in the running text as OMP"

I. 79-88 quotations and **Supplementary Table 3** are given in the response to comment 3.

20. I. 182: Does it mean that the value $F_l = 76 \pm 12$ nmol l⁻¹d⁻¹, found for August is used for the whole stratified period? Or that the same calculation is done for each weekly measurement?

Response: The lateral methane input F_l is based on methane measurements in the experimental enclosures (4 repetitions in enclosures 1 and 5 repetitions in enclosure 13) and the open lake (4 repetitions) sampled during the first two weeks in Aug 2014. This value has been used to parameterize the mass balance for the stratified period. We have now clarified this in the text (I. 206-207, I. 511-514).

I. 206-207: "...using as F_l term (lateral methane input) the value obtained for August 2014 as described above."

I. 511-514: “...methane measurements were taken inside mesocosm enclosures (2 weeks after the water was exchanged with open lake water) and in the open water adjacent to the enclosures in the South basin (details in Supplementary Table 3)”

21. I. 188 to 193: The Monte Carlo simulations result in a large standard deviation. Is the SD reported 1sigma? 2sigma? Which confidence percentiles? The SDs are as large as or larger than the derived value. Could the authors comment on these large uncertainties? Similarity to previous studies? Relevance of results?

Table1: give the definition of the SD

Response: Standard deviation for surface emission/lateral sediment input/ diffusion from thermocline was computed as $\sqrt{[\sum(x - \bar{x})^2]/(n-1)}$. The Monte Carlo simulation was performed in R deploying the rnorm-function. Normal distribution was characterised with the density equation $f(x) = 1/(\sqrt{2\pi}\sigma) e^{-((x - \mu)^2/(2\sigma^2))}$. We have added these information into the method section (under Monte Carlo simulation). Furthermore, the foot notes of Table 1 now state that Monte Carlo results are given as mean 1*SD notation. We have also included the resulting density curves with additional information in Supplementary Fig. 5:

Supplementary Figure 5 | Density curve of oxic methane production rates obtained from mass balancing. Monte Carlo simulation was deployed to solve methane mass balances (9999 iterations). The density function was computed as $f(x) = 1/(\sqrt{2\pi}\sigma) e^{-((x - \mu)^2/(2\sigma^2))}$. Here σ is the standard deviation and μ is the mean value.

For the mass balance analysis, OMP rates are deduced from the differentials between methane gain and loss terms and the corresponding parametrization. Negative production rates indicate underestimated methane oxidation (which can exceed OMP rates) or overestimated anoxic methane input instead of “uncertainty”. In fact, it is possible that positive OMP rates were masked by strong methane oxidation rates leading to negative mass balance output. This is a direct result of a conservative Monte Carlo configuration and underlines that OMP is an important contributor to surface emission.

As described throughout the sensitivity analysis, OMP accounts for a substantial part of methane surface emission in Lake Stechlin. Even at much lower OMP rates, the contribution still remains important (see sensitivity analysis throughout the discussion, for instance, I. 319-322, I. 332-335). So far, only 3 OMP rate estimations are available in the literature: Lake Stechlin (up to 58 nmol l⁻¹ d⁻¹), Lake Hallwil (76-138 nmol l⁻¹ d⁻¹) and Lake Cromwell (230±10 nmol l⁻¹ d⁻¹). Our mass balance approach gives OMP rates in the same range.

The corresponding oxic methane production rates are now placed into literature context throughout the discussion, for instance, in lines 363-383.

I. 319-322: “Applying these alternative emission rates to the mass balance analysis gave an OMP rate between 41 and 185 nmol l⁻¹ d⁻¹, which still translated to a substantial oxic methane contribution (32 – 68 %) to the surface methane emission...”

I. 332-335: “Even doubling the lateral methane input — an unlikely scenario for a meso-oligotrophic lake such as Lake Stechlin— still could not fully explain the observed SML methane in the Northeast basin, and a substantial OMP rate (19 nmol l⁻¹ d⁻¹) would still be required to balance the methane budget”

I. 363-383: “Comparing our measurements and assumptions against literature values shows that our mass balance analysis is reasonably parametrized and robust. The system-wide methane emission from the SML in the Northeast basin was estimated to be 942 mol d⁻¹ in the stratified period, of which 32 % from lateral input (372 mol d⁻¹) and 5 % from vertical diffusion from the thermocline (56 mol d⁻¹) (Table 1). Similarly, methane emission from the SML in the South basin was 795 mol d⁻¹, and only 45 % (423 mol d⁻¹) could be attributed to lateral input and 4 % (41 mol d⁻¹) to vertical input from the thermocline. The deficits (plus additional consumption via methanotrophy), therefore, must be compensated for by internal oxic methane production. The estimated OMP rate averaged over the stratified period was (mean±SD) 71±74 nmol l⁻¹ d⁻¹ (Northeast basin) and 88±75 nmol l⁻¹ d⁻¹ (South basin). An earlier study¹⁵ using bottle incubations measured a net OMP rate of up to 58 nmol l⁻¹ d⁻¹ for Lake Stechlin, which corresponds to a hypothetical gross production rate of 75 nmol l⁻¹ d⁻¹ when assuming 30 % oxidation. Similar OMP rates have also been estimated for Lake Hallwil, between 76 and 138 nmol l⁻¹ d⁻¹²¹ (Supplementary Note 1). Particularly high OMP values, such as what we found in late June (mean±SD; 236±32 nmol l⁻¹ d⁻¹), have also been reported by others³² (e.g. 230±10 nmol l⁻¹ d⁻¹ in Lake Cromwell, Canada).

Overall, by accounting for the different methane sources and sinks in the SML mass balance analysis, we show that OMP is a key contributor to system-wide surface emission in Lake Stechlin. This conclusion is consistent with previously reported OMP rates obtained from bottle incubations¹⁵ and is not sensitive to inherent uncertainties in our mass balance approach as shown by the sensitivity analysis.”

DISCUSSION.

22. I. 223: Rephrase “3) the substantial contribution ... methane emission (64%..)

Response: We have rephrased it from ‘the contribution of OMP to Lake Stechlin’s system-wide methane emission was substantial (64 % in the Northeast basin and 50 % in the South basin)’ to ‘OMP contributed substantially to the system-wide methane emission of Lake Stechlin’s Northeast (64 %) and South basin (50 %)’ (I. 292-295).

23. I. 226 to 230. The model discussed here has not been mentioned previously nor its results. More details should be given to strengthen this statement.

Response: We have moved the presentation of the model to the result section (I. 241-280) and added more explanation throughout the discussion (e.g. I. 401-406, I. 411-416, I. 425-427).

I. 401-406: “The system-wide contribution of the anoxic methane sources is mainly controlled by littoral sediment flux and the corresponding littoral sediment area. Trophic state^{52,53} and temperature^{12,54} are important drivers of the methane flux from sediments. Higher sediment methane fluxes in eutrophic systems and in warmer climate zones compared to our dataset of stratified meso-to-oligotrophic lakes in the temperate region could shift the curve of the empirical models to the right (Fig. 4, Supplementary Fig. 6).”

I. 411-416: “The OMC prediction, therefore, may vary mainly for small lakes which have been reported to cause less methane emission on a global scale compared to large lakes²⁸ (<0.01 versus >1 km²). It shall be noted that the model predictions based on A_{sed} and V will be more reliable than based exclusively on lake surface area due to sediment steepness/aspect ratio and total depth modulating the littoral sediment area at constant lake surface area.”

I. 425-427: “By increasing data resolution in our empirical models, the models can then be used to further improve the global methane emission assessments.”

24. I. 240: Please specify the previously reported range. Does the SD measured in this study fit the range?

Response: We specified the previously reported range for Stechlin (2.6 mmol m⁻² d⁻¹ ±42 %; McGinnis et al. 2015). Also, we compared our values to the global mean estimates for this region. Reported standard deviations for some other lakes exceeded 100 % of the mean value (Sabrekov et al. 2017, Xiao et al. 2017), in comparison the standard deviation of Stechlin’s surface emission was 34 % (South basin) or 57 % (Northeast basin), respectively. Additionally, we highlighted that our k_{600} -wind relationship is very similar to the relationship developed in Lake Hallwil (Donis et al. 2017) (I. 313-317).

We have made it clear that observed values fit the range of earlier reports (I. 305-313). Note, strong variability in OMP rates in surface waters may substantially contribute to emission heterogeneity.

I. 306-314: “The average surface methane emission during the stratified period was 0.47 mmol m⁻² d⁻¹ (±57 % SD) in the Northeast basin and 0.71 mmol m⁻² d⁻¹ (±34 % SD) in the South basin (taken mainly during calm weather). The larger value in the South basin can be attributed to higher influence from littoral methane sources. However, these emission values are comparable with the global estimate of 0.62 mmol m⁻² d⁻¹ for the region 25-54° latitude⁴² and within the range reported earlier for Lake Stechlin⁴³ (exceeding 4 mmol m⁻² d⁻¹ at strong wind; 2.6 mmol m⁻² d⁻¹ ±42 % SD). Highly variable surface emission has been reported earlier, for some systems standard deviations exceed 100 % of mean emission values during summer^{24,26}.”

I. 313-317: “In case of the South basin we estimated the emission from wind speed data and the corresponding results are dependent on the gas transfer constant (k_{600}) value used. Our k_{600} -wind speed relationship ($k_{600}[\text{cm h}^{-1}] = 1.98 \cdot U_{10}[\text{m s}^{-1}] + 0.98$) was very similar to an earlier report (e.g. Lake Hallwil: $k_{600}[\text{cm h}^{-1}] = 2.0 \cdot U_{10}[\text{m s}^{-1}]$; Donis et al. ²¹).”

McGinnis et al. 2015, *Environ. Sci. Technol.* **49**, 873-880

Sabrekov et al. 2017, *Biogeosciences* **14**, 3715-3742

Xiao et al. 2017, *J. Geophys. Res. Biogeosci.* **122**, 1597-1614

Donis et al. 2017, *Nat. Commun.* **8**, 1661

25. l. 242-243: what is the point here?

Response: We apologise for not making this clearer. The surface emission was determined by two different methods between the Northeast basin and the South basin (see Methods); nevertheless, both methods lead to the same conclusion that OMP is a major contributor of surface methane emission. Therefore, the overall conclusion appears to be robust regardless of the methods used. We have clarified this point in the text (l. 322-324).

l. 322-324: “In other words, regardless of the method or model used to estimate surface methane emission, it remains that OMP was an important contributor to surface emission.”

26. l. 244: could the authors comment on the differences between the two basins?

Response: Please see our response to comment 17. DelSontro et al. showed that lateral methane transport from shore to the basin-centre follows a predictive function, the shorter the distance, the higher the resulting accumulation in the mid-water column. Stechlin’s South basin is about half the size of the Northeast basin but accounts for ca. the same amount of littoral sediment area (the main source of anoxic methane). This morphological difference causes more littoral methane to reach mid-waters and followingly increases mid-water surface emission. We have now highlighted the morphology difference of the two basins in the result section (l. 251-253).

l. 251-253: “While Stechlin’s Northeast and South basins vary in surface area (NE: 2.01 km²; S: 1.12 km²) and SML volume V (NE: 11,200,000 m³; S: 5,700,000 m³), their littoral sediment areas are comparable (NE: 0.28 km², S: 0.31 km²) (values given for a 6 m deep SML).”

DelSontro et al. 2018, *Ecosystems* **21**, 1073-1087

27. l. 248: refer to Table S5.

Response: The original manuscript version already included a reference to Supplementary Table 5 at the end of the very same sentence (Supplementary Table 7 in the new version) (l. 319-322).

l. 319-322: “Applying these alternative emission rates to the mass balance analysis gave an OMP rate between 41 and 185 nmol l⁻¹ d⁻¹, which still translated to a substantial oxic methane contribution (32 – 68 %) to the surface methane emission (details in Supplementary Table 7).”

28. l. 254: Specify the previously reported range.

Response: We have added the corresponding literature values to the text (l. 327-332).

l. 327-332: “It is within the range of fluxes reported for other temperate water bodies (e.g. Rzeszów Reservoir, Poland⁴⁴: mean±SD 0.69±0.56 mmol m⁻² d⁻¹ in May-Sep; Lake Hallwil, Switzerland²¹: 1.75±0.2 mmol m⁻² d⁻¹ in Sep (Supplementary Note 1); Boltzmann-Arrhenius equation at ca. 20°C¹²: ca. 2 mmol m⁻² d⁻¹, including Lake Constance (Überlingen basin)/ Lake Ammer/ Lake Königsegg/Reservoir Schwarzbach in Germany¹² with ca. 1.3 mmol m⁻² d⁻¹).”

29. l.357-358 and l. 379: Recall 1) the large uncertainty and lack of knowledge on the processes and transport pathways 2) the uncertainty in individual lake measurements and issues related to upscaling (definition of lakes, ponds, reservoirs, uncertainty on surface areas...). A better understanding of the processes and transport pathways may allow a better modelling approaches to derive other estimates of methane emissions from lakes, and avoid issues of upscaling, as currently done. This might help for a better assessment of methane emissions from lakes and reservoirs,

though the values derived here for a single lake, are highly uncertain. More comment and discussion on the global perspective and methane budget is needed to highlight the added value of defining novel predictive models.

Response: As suggested by the reviewer, we have addressed the uncertainty in global assessments and the added value of our study in the subsection of the discussion ‘Global implications’ (l. 385-393). As described earlier, we have moved the presentation of the empirical model to the result section and added more information throughout the discussion (e.g. l. 401-406, l. 410-415, l. 425-427).

l. 385-393: “In addition to known knowledge gaps in the global methane dynamics^{22,23}, OMP has not been considered as source of uncertainty in global assessments^{1,2,22,23}. Because both oxic and anoxic methane sources in lakes can be modulated by multiple factors and processes (Supplementary Fig. 7), some of which are still poorly understood, it would be premature to construct a mechanistic model to fully describe methane dynamics in lakes. Instead, we developed empirical models as useful tools to predict the contribution of OMP to the system-wide emission (OMC) in stratified meso-to-oligotrophic lakes in the temperate region based on a set of simple lake morphological parameters (Fig. 4, Supplementary Fig. 6).”

l. 401-406: “The system-wide contribution of the anoxic methane sources is mainly controlled by littoral sediment flux and the corresponding littoral sediment area. Trophic state^{52,53} and temperature^{12,54} are important drivers of the methane flux from sediments. Higher sediment methane fluxes in eutrophic systems and in warmer climate zones compared to our dataset of stratified meso-to-oligotrophic lakes in the temperate region could shift the curve of the empirical models to the right (Fig. 4, Supplementary Fig. 6).”

l. 410-415: “The OMC prediction, therefore, may vary mainly for small lakes which have been reported to cause less methane emission on a global scale compared to large lakes²⁸ (<0.01 versus >1 km²). It shall be noted that the model predictions based on A_{sed} and ∇ will be more reliable than based exclusively on lake surface area due to sediment steepness/aspect ratio and total depth modulating the littoral sediment area at constant lake surface area.”

l. 425-427: “By increasing data resolution in our empirical models, the models can then be used to further improve the global methane emission assessments.”

2) Reviewer #2 (Remarks to the Author):

1. This paper reinforces the importance of oxic methane in lake methane emission, and analyzes the contribution of oxic methane production to surface methane emission in two portions of a temperate lake, and with a hope to project its global contributions with additional few lakes in the literature. The main contribution of this manuscript would be on its global context.

Response: Many thanks for the constructive comments. We hope that all comments have been addressed to full satisfaction. Please note that we have made changes to the introduction and discussion sections to highlight the value of this study for global methane emission assessments. Also, the presentation of the empirical model has been moved to the result section. We have also changed the order of references and supplementary figures and tables.

2. This paper is well written.

Response: Thank you for the compliment.

3. Lake Stechlin does not have major inflows or outflows. However, most freshwater lakes globally have flows. The representativeness of this lake is low in the global context.

Response: The effect of river in- and outflow is considered by the mass balance approach; please see Equation 1 in the method section: The term $(Q_R * C_R)$ represents inflow and term $(Q_R * C_R)$ the outflow. Oxic methane production is an additional methane source in lakes beside the anoxic source and it increases the methane emission. Our empirical model includes lakes with river in-/outflow e.g. Lake Hallwil, St. Jean, Lake Simard and Lake Camichagama. Our results indicate that regardless of river connections, oxic methane production is an important contributor to atmospheric emission.

Ground water flow into the lake can modulate the sediment-to-water methane flux (e.g. increased sediment methanogenesis due to more organic carbon is available). This effect is captured by our mass balance approach as we determine the sediment-to-water methane flux by comparing enclosure with open-lake measurements.

Further, we want to emphasize that while there has been increasing evidence of OMP in aquatic environments, its contribution to global methane emission has not been considered in depth. Our study is to our knowledge the first attempt to evaluate the global relevance of OMP, especially in large lakes. Our empirical model shows that the contribution of oxic methane sources to the system-wide atmospheric emission follows a specific relationship to morphological parameters (littoral sediment area, surface mixed layer volume, lake surface area) of the lakes. Applying this empirical model to global lake data, we arrived at the conclusion that OMP is a significant part of the global methane emission assessment.

4. Two basins? How there are many ways of cutting portions. The separation might not affect the mass balancing model but would change the global prediction relationship. In general, I do not agree using the two portions of the same lake as two individual lakes in the subsequent analysis.

Response: Hofmann et al. 2010 showed that methane produced in the littoral zone is a major source of mid-water methane. Building upon this study, DelSontro et al. 2018 demonstrated that the corresponding lateral transport is a function of the shore-centre-distance. Stechlin's South basin has a surface area of 1.12 km² and a littoral sediment area of 0.31 km². In contrast, the Northeast basin

is scaled to 2.01 km² surface area and 0.28 km² littoral sediment area. At about half the size but the same littoral sediment area, more littoral methane is reaching the South basin's mid-water column compared to the Northeast basin. This agrees with recorded methane profiles (Fig. 2) and recorded surface emissions (Supplementary Table 4): Higher methane accumulation and surface emission was measured in the South basin. We want to emphasize, that we investigated the predictive power of morphological parameters. Due to the same geological history, two basins of the same lake will have similar biochemical properties. As shown by the mass balances, the morphological parameters lead to considerable differences between the two basins. That is why we considered the two basins separately.

Additionally, we followed the reviewer's idea considering all basins combined. The mass balance was not affected (Table 1 legend). When the results were incorporated into our predictive model (Fig. 4), the whole-lake data point fell on the same trend line (see open square symbol in Fig. 4) reinforcing the explanatory power of our model. Note, the alternative regression results are stated in the figure legend (even slightly better).

Table 1 | Mass balance components for estimating the oxic methane source. Oxic production was computed by measuring/estimating surface emission, oxidation, lateral input, as well as vertical diffusion (see Fig. 1) and solving the mass balance for the missing component.

Site	Mass Balance Component	Symbol	Whole System		Per Volume
			[mol d ⁻¹]	[kg d ⁻¹]	[nmol l ⁻¹ d ⁻¹]
Northeast basin	Surface emission	F_S	942±538	15±9	90±52
	Methane oxidation	MOx	226	4	22
	Lateral sediment input	F_L	372±57	6±1	36±6
	Diffusion from thermocline	F_z	56±55	1±1	5±5
	Internal (oxic) production	P_{net}	752±771	12±12	72±74
South basin	Surface emission	F_S	795±268	13±4	148±50
	Methane oxidation	MOx	141	2	26
	Lateral sediment input	F_L	423±65	7±1	79±12
	Diffusion from thermocline	F_z	41±54	1±1	8±10
	Internal (oxic) production	P_{net}	470±400	8±6	88±75

Seven replicate measurements were taken in the open water of the Northeast (69.5 m deep; surface area 2,006,700 m²; 53°09'20.2"N 13°01'51.5"E) and South basin (20.5 m deep; surface area 1,122,775 m²; 53°08'36.6"N 13°01'42.8"E) of Lake Stechlin during the stratified period in 2016 (June-July). Values listed as mean±SD. Note that Monte Carlo simulation was used to solve the mass balance after the target component (in bold; mean±1*SD) (see methods for details). Supplementary Fig. 5 illustrates the density function of the Northeast and South basin dataset. If the Monte Carlo simulation were to be applied to whole lake data (combining South and Northeast basins data), oxic methane production rates P_{net} do not change: 78±80 nmol l⁻¹ d⁻¹ ($F_S = 2503±1160$, MOx = 496, $F_L = 1198±185$, $F_z = 139±170$, $P_{net} = 1653±1703$ mol d⁻¹).

Figure 4 | Oxidic methane contribution (OMC) in relation with lake morphology of stratified meso-to-oligotrophic lakes in the temperate region. Illustration of Lake Hallwil (McGinnis et al.⁴⁰; modified), Stechlin (bathymetry data) and Cromwell ([https://crelaurentides.org/dossiers/eau-lacs/atlasdeslacs?lac=11935#photo\[1193%5\]/0/](https://crelaurentides.org/dossiers/eau-lacs/atlasdeslacs?lac=11935#photo[1193%5]/0/); modified). The ratio of sediment area (A_{sed}) and SML volume (V) determines the OMC. The trend line (red line) follows the exponential function $y = 87.49e^{-7.61x}$ ($R^2 = 0.95$, $p \ll 0.01$, Std.error = 8.6 %). The y-axis is scaled to $\log_{2.7}$ and the x-axis is linear. With increasing lake size, V increases quicker than A_{sed} making OMC the largest source of SML methane in lakes with $A_{sed}/V \leq 0.07 \text{ m}^2/\text{m}^3$. Lake Hallwil estimation²¹ was updated as described in Supplementary Note 1; the lower and upper end (error bars) were used to compute the mean OMC which was used for developing the trend line function. Estimations for other lakes were computed as defined in Supplementary Note 3. If whole lake data (combining South and Northeast basin data) was to be applied to this empirical model (empty symbol) the regression constants and statistics only changes minimally ($y = 88.48e^{-7.56x}$; $R^2 = 0.96$, $p \ll 0.01$). Source data are provided as a Source Data file.

Hofmann et al. 2010, *Limnol. Oceanogr.* **55**, 1990-2000

DelSontro et al. 2018, *Ecosystems* **21**, 1073-1087

5. Line 144. Surface methane emissions other than the Northeast portion were derived from wind speed, based on a linear relationship, but not so strong ($r^2 = 0.44$). How would this impact on the balancing analysis?

Response: The R^2 value of regressing k_{600} data over wind speed is no measure for applicability. Our dataset includes seasonal emission data where natural variation is expected due to changing conditions in turbulence (MacIntyre et al. 2010), heat flux (MacIntyre et al. 2010), effective wind forcing (Vachon and Prairie 2013), precipitation (Ho et al. 2007), etc.

To assess uncertainty during modelling methane emission, we compared our flux model to 6 alternative emission models taken from the literature (Donis et al. 2017, MacIntyre et al. 2010, Vachon and Prairie 2013) that consider additional parameters (e.g. lake area, heat flux). Those more complex models gave a range of values of which our model was close to the median value (Supplementary Table 7), suggesting that our model is reliable. We further evaluated the potential impact of variability by doing a sensitivity analysis (see ‘Local implications’ in the discussion).

Regardless which model is used, the overall conclusion remains that OMP is a major methane source in Lake Stechlin. Please note that the alternative method of flux chamber measurements used in the Northeast basin leads to the same conclusion.

We included the information that literature models were used to validate modelled emission values for the South basin (l. 169-171).

l. 169-171: “Other published models^{21,37,38} in the literature (mainly based on direct turbulence measurements)^{37,38} were used to validate these emission values (see sensitivity analysis in Discussion).”

MacIntyre et al. 2010, *Geophys. Res. Lett.* **37**, L24604

Vachon and Prairie 2013, *Can. J. Fish. Aquat. Sci.* **70**, 1757-1764

Ho et al. 2007, *J. Marine Syst.* **66**, 150-160

6. Fig. 3. Methane concentration jumps quickly in early stratified state and drops back quickly in the rest of the state. However, the methane measurements were made in the early state not throughout the entire stratified state. This could lead to substantial over-estimation on OMP and bias in the analysis.

Response: We would like to point out that Figure 3 shows oxic methane production rates.

The production rates observed in our study are within the range of earlier reports (Grossart et al. 2011, Bogard et al. 2014, Donis et al. 2017), this includes the highest observed average OMP rate of $236 \pm 32 \text{ nmol l}^{-1} \text{ d}^{-1}$. Furthermore, in a parallel study (manuscript submitted), some authors of this study observed a similarly high OMP rate in the same months of year (in 2017) suggesting that the high seasonal production rate is a recurring phenomenon. As OMP has been associated with phytoplankton (Bizic-Ionescu et al. 2019, Lenhart et al. 2016), variability in production rates is expected. The average production rates presented throughout the manuscript are ca. $72\text{-}88 \text{ nmol l}^{-1} \text{ d}^{-1}$ for both basins which is in the range of the other data points in Fig. 3. Please note, the yellow data point presented in Figure 3 (August value, ca. $100 \text{ nmol l}^{-1} \text{ d}^{-1}$) represents measurements in the enclosures that were filled with lake water prior to the measurements. We used enclosure measurements and compared them to open-lake measurements (South basin) to obtain the lateral input. Therefore, the yellow data point includes multiple open-lake data from August 2014; we did not state this fact as we wanted to separate the seasonal development from estimating the lateral input.

As indicated by water column profiles in Lake Stechlin taken throughout the full stratified season including August and September (Bizic-Ionescu et al. 2018), comparable methane conditions are observed at the end of the stratified season. Please note, that the OMP estimate for Lake Hallwil (Donis et al. 2017) includes the highest methane content in August and September.

Bogard et al. 2014, *Nat. Commun.* **5**, 5350

Bizic-Ionescu et al. 2018, Springer International Publishing AG, A. J. M. Stams, D. Z. Sousa (eds.), Biogenesis of Hydrocarbons, Handbook of Hydrocarbon and Lipid Microbiology

Donis et al. 2017, *Nat. Commun.* **8**, 1661

Grossart et al. 2011, *PNAS* **108**, 19657-19661

Lenhart et al. 2016, *Biogeosciences* **13**, 3163-3174

7. Lines 188-192. The OMP rates at 72 ± 74 and 88 ± 75 are of very high uncertainty. How much is the uncertainty to the 64% and 50%?

Response: We used Monte Carlo simulation to obtain an average value for oxic methane production rates. The uncertainty caused by the variability of individual mass balance components are discussed throughout a sensitivity analysis presented in the discussion section under “Local implications” (e.g. l. 319-322, l. 333-336, l. 359-362).

The mass balance analyses computed oxic methane production rates with considerable standard deviation which might be attributed to 1) seasonal variability in individual mass balance components and 2) intermittency of oxic methane production due to the association with primary production (Grossart et al. 2011, Bogard et al. 2014) and phytoplankton (Lenhart et al. 2016, Yao et al. 2016). Researching the intermittent character of the oxic methane source is a great opportunity to understand better and potentially reduce the global upscaling uncertainty (e.g. 100 % minimum uncertainty range in the latest global methane budget, Saunois et al. 2016)

We added the missing reference of the sensitivity analysis to the revised manuscript version (l. 217-218).

l. 319-322: “Applying these alternative emission rates to the mass balance analysis gave an OMP rate between 41 and 185 $\text{nmol l}^{-1} \text{d}^{-1}$, which still translated to a substantial oxic methane contribution (32 – 68 %) to the surface methane emission (details in Supplementary Table 7)”

l. 333-336: “Even doubling the lateral methane input — an unlikely scenario for a meso-oligotrophic lake such as Lake Stechlin— still could not fully explain the observed SML methane in the Northeast basin, and a substantial OMP rate ($19 \text{ nmol l}^{-1} \text{d}^{-1}$) would still be required to balance the methane budget.”

l. 359-362: “If we consider the extreme scenario by completely ignoring methane oxidation ($\text{MOx} = 0$), the estimated average OMP rate for the South basin would decrease to (mean \pm SD) $40 \pm 53 \text{ nmol l}^{-1} \text{d}^{-1}$ and would still remain an important SML methane source (32 %).”

l. 217-218: “A sensitivity analysis (see discussion) examined the effect of variable mass balance components on the contribution pattern.”

Bogard et al. 2014, *Nat. Commun.* **5**, 5350

Grossart et al. 2011, *PNAS* **108**, 19657-19661

Lenhart et al. 2016, *Biogeosciences* **13**, 3163-3174

Saunois et al. *Earth Syst. Sci. Data* **8**, 697-751

Yao et al. 2016, *Appl. Environ. Microbiol.* **82**, 6994-7003

8. Line 277. How many lakes were evaluated for the range between 4 and 30 $\text{nmol l}^{-1} \text{d}^{-1}$? What are these lakes’ characteristics?

Response: We used the methane oxidation (MOx) rates reported by Bogard et al. 2014 (Lake Cromwell) and Donis et al. 2017 (Lake Hallwil) and Oswald et al. 2015 (Lake Rotsee). The original manuscript version included references to these studies (l. 277 original version, l. 353-354 in the revised manuscript version).

Lake	Size	Mixed/stratified	Region	Trophy
------	------	------------------	--------	--------

Lake Cromwell, Canada	0.1 km ²	Stratified	Temperate	Mesotrophic
Lake Rotsee, Switzerland	0.5 km ²	Stratified	Temperate	Eutrophic
Lake Hallwil, Switzerland	10 km ²	Stratified	Temperate	Mesotrophic

In comparison, the South (Northeast) basin of Stechlin is 1 (2) km² large, stratified, in the temperate region and at the border to mesotrophic nutrient state.

It should be noted that MOx at the sediment-water interphase is accounted for as we estimated the net methane flux released from littoral sediments into water column. Furthermore, MOx is a loss term in the mass balance; applying higher MOx rates will translate to higher oxic methane production rates. We parametrized the mass balance conservatively and considered the extreme scenario when MOx is absent (most conservative consideration possible; l. 359-362).

l. 353-354: “...MOx rates in oxic surface waters have been reported to range between 4 and 30 nmol l⁻¹ d⁻¹ 21,32,51”

l. 359-362: “If we consider the extreme scenario by completely ignoring methane oxidation (MOx = 0), the estimated average OMP rate for the South basin would decrease to (mean±SD) 40±53 nmol l⁻¹ d⁻¹ and would still remain an important SML methane source (32 %).”

Bogard et al. 2014, *Nat. Commun.* **5**, 5350

Donis et al. 2017, *Nat. Commun.* **8**, 1661

Oswald et al. 2015, *PLOS ONE* **10**, e0132574

9. Line 278. “MOx to be equivalent to a constant fraction (30 %) of the internal production”. This arbitrary treatment would need a justification.

Response: Methane oxidation (MOx) can vary depending on substrate availability, product pool, microorganism and environmental settings. In a complementary study, some of us (in collaboration with a different group of researchers) used isotope labelling techniques and measured an MOx of ca. 21% (fraction 0.21) of oxic methane production rates (in September when water column methane concentration was lower than in June/July). Those data, however, cannot be included here because of different data ownership. Using a constant fraction of 0.3, our mass balance analysis yields rather conservative OMP values, but agree with earlier measurements (Grossart et al. 2011) and therefore we consider it suitable for budget calculations. More importantly, as MOx is a loss term in the methane mass balance, using a higher MOx value would only increase the estimated OMP rates (and OMC). In the manuscript, we considered the extreme scenario when MOx is absent (hence the most conservative OMP estimate) and the mass balance result showed that the oxic source was still a substantial contribution to methane emission.

We now included a reference to the method section (l. 354-356).

l. 355-357: “For our study, we assumed MOx to be equivalent to a constant fraction (30 %) of the internal production during the stratified season (see method section for details).”

Tang et al. 2014, *Limnol. Oceanogr.* **59**, 275-284

Grossart et al. 2011, *PNAS* **108**, 19657-19661

10. Again, there are too many simplified treatments lack of reasonable justifications.

Response: It is not clear what this comment refers to. We assume the reviewer refers to the comment 9 or 11.

11. In the global prediction, it is inappropriate to use the two portions of the lake as two lakes in the predictive model. Even the same lake has very different methane concentration, indicating lake size is not sufficient. There are many other factors to be considered such as flow conditions, wind fields, carbon richness in lakebed, lake shape and depth, and so on, in addition to size.

Response (*'In the global prediction, it is inappropriate to use the two portions of the lake as two lakes in the predictive model. Even the same lake has very different methane concentration, indicating lake size is not sufficient.'*): (Please also see our response to comment 4) Earlier studies showed that lateral transport of methane from littoral zones is an important source of mid-water methane in the surface mixed layer (Hofmann et al. 2010, DelSontro et al. 2018), and its distribution follows a predictive function (DelSontro et al. 2018): The higher the shore-to-centre distance the less methane reaches the middle of the lake; at distances above ca. 2 km no littoral methane reaches the mid-water column. Therefore, morphological parameters have been shown earlier as good predictive parameters for mid-water column methane profiles and emission patterns. For example, Stechlin's South and Northeast basin differ in surface area (1.12 km² vs. 2.01 km²) but their littoral sediment areas are comparable (0.31 km² vs. 0.28 km²). Comparable sediment methane input distributed over half the lake size leads to higher water column methane profiles and surface emission.

We reanalysed the empirical model using whole-lake data instead of separated basins data. The resulting data point falls on the same trend line (open square symbol in Fig. 4). This outcome reinforces the explanatory power of our predictive model. We have added the alternative regression results to the figure legend (l. 278-280).

Fig. 4 and corresponding legend are displayed in the response to comment 4.

Response (*'There are many other factors to be considered such as flow conditions, wind fields, carbon richness in lakebed, lake shape and depth, and so on, in addition to size.'*): Our empirical models are not meant to represent the full biological and physical complexity and intricacy of the methane cycle, and we do not deny that there can be many more factors that affect methane dynamics in the environment. It would be impractical, if not impossible, to try and develop a complete mechanistic model when scientists do not even know all the factors involved in methane production and consumption. (Case in point: the omission of oxic methane production in IPCC reports) However, the purpose of empirical models is to reduce a system of intractable complexity to a manageable set of simple parameters for making useful predictions.

Our models are useful tools to predict the contribution of the oxic methane source to the surface emission. Further, our model is based on stratified, meso-to-oligotrophic lakes in the temperate region. We clearly stated throughout the discussion that variability of these parameters can alter the trendline function. Also, our dataset includes lakes with and without river connection (see also our response to comment 3), though our model explains nearly all variance in the data (Fig. 4). We are not opposed to more sophisticated models if and when such models are called for.

l. 278-280: "If whole lake data (combining South and Northeast basin data) was to be applied to this empirical model (empty symbol) the regression constants and statistics only changes minimally ($y = 88.48e^{-7.56}$; $R^2 = 0.96$, $p \ll 0.01$)."

12. The global extrapolation is projected from only a few lakes found in the literature. However, their representativeness to global lakes is not discussed. Large variations many found among these selected lakes. We can imagine the huge discrepancy of lakes across the Earth, given the wide variety of landscape, lake history, flow connections, etc. This over-simplified extrapolation is questionable and unconvincing.

Response: While there has been increasing evidence of OMP in aquatic environments, its contribution to global methane emission has not been considered in depth. Our study is to our knowledge the first attempt to evaluate the global relevance of OMP. We do not pretend that our models capture all of the biological, physical and geological differences among lakes across the Earth. Rather, the purpose of our empirical models is to reduce a system of intractable complexity to a manageable set of simple parameters for making useful predictions. We took a conservative approach to our predictions (e.g. l. 529-530 or using mean values instead of upper-end estimations for oxic methane production rates in Lake Stechlin/Lake Hallwil, Lake size class areas; see Fig. 4 and Supplementary Note 4) and we included a sensitivity analysis to evaluate the impacts of variability on our conclusions (see ‘Local implications’ in the discussion section, e.g. l. 332-335, l. 359-362). Applying our empirical models to global lake data, we arrived at a new understanding that oxic methane sources can be an important contributor to methane emissions.

In the revised manuscript we acknowledged the complexity of methane cycling in lakes (see Supplementary Fig. 7), as well as that increased data resolution in the empirical model will improve the global emission assessments (l. 422-427).

Note, we have changed the manuscript title from ‘Contribution of oxic methane production to surface methane emission in lakes – local and global **importance**’ to ‘Contribution of oxic methane production to surface methane emission in lakes – local and global **implication**’.

l. 332-335: “Even doubling the lateral methane input — an unlikely scenario for a meso-oligotrophic lake such as Lake Stechlin— still could not fully explain the observed SML methane in the Northeast basin, and a substantial OMP rate ($19 \text{ nmol l}^{-1} \text{ d}^{-1}$) would still be required to balance the methane budget.”

l. 359-362: “If we consider the extreme scenario by completely ignoring methane oxidation ($\text{MOx} = 0$), the estimated average OMP rate for the South basin would decrease to (mean \pm SD) $40\pm 53 \text{ nmol l}^{-1} \text{ d}^{-1}$ and would still remain an important SML methane source (32 %).”

l. 529-530: “To obtain a conservative estimate of OMP in the SML, maximum K_z values within the bottom 3 m of the SML were used to compute F_z .”

l. 422-427: “Such a surprising finding justifies the need for further investigation of OMP in lakes world-wide with different geological histories, trophic states, climates, and physical (e.g. lake colour, stratification patterns or with strong in-/ out flow) and chemical characteristics (e.g. alkaline versus acidic) (Supplementary Fig. 7). By increasing data resolution in our empirical models, the models can then be used to further improve the global methane emission assessments.”

Supplementary Fig. 7 is displayed on the next page.

Supplementary Figure 7 | Examples of factors affecting the contribution of oxic and anoxic methane sources to the system-wide surface emission. Factors are categorized into morphology, sediment characteristics, nutrient conditions/ecology, meteorology and lake physics. A_{sed} symbolizes littoral sediment area, SML is surface mixed layer, V refers to the volume of the surface mixed layer and Q is flow rate.

13. Line 66. After this study, it still does not seem to be clear on “what drives the underlying processes and what causes different contribution patterns in different lakes.” Otherwise, the global predictive model would be more sophisticated.

Response: We developed predictive models building upon earlier studies (Hofmann et al. 2010, Grossart et al. 2011, Bogard et al. 2014, Donsi et al. 2017, DelSontro et al. 2018). Morphological parameters explained nearly all variance in our models. Our aim was not to develop mechanistic models that would include all ‘underlying processes and causes’. In fact, we believe such a mechanistic model is only attainable when all the processes and factors behind methane production and consumption are fully known. We would re-iterate the fact that oxidic methane production is only a recent discovery and the underlying biochemical pathways are still being studied. Also, our empirical models are not meant to represent the full complexity and intricacy of the ecosystem but rather are useful tools for making predictions based on simple, measurable parameters. We are not opposed to more sophisticated models if and when such models are called for.

We have rephrased the corresponding sentence (l. 71-74).

l. 71-74: “While both studies demonstrate that OMP can be an important source of methane emission, it is not clear if OMP is a general phenomenon in lakes and what may explain the different contribution patterns in different lakes.”

Hofmann et al. 2010, *Limnol. Oceanogr.* **55**, 1990-2000

Grossart et al. 2011, *PNAS* **108**, 19657-19661

Bogard et al. 2014, *Nat. Commun.* **5**, 5350

Donsi et al. 2017, *Nat. Commun.* **8**, 1661

DelSontro et al. 2018, *Ecosystems* **21**, 1073-1087

REVIEWERS' COMMENTS:

Reviewer #1 (Remarks to the Author):

The authors present estimates of oxic methane production in lake and its contribution to surface to atmosphere flux, using a mass balance analysis based on in-situ measurements. The measurements have been performed all year round (allowing seasonal variation study) and in two basins of one lake. Complementary analysis has been performed in an artificial laboratory lake. In their study the authors estimate that oxic methane production is an important contribution to surface-water emissions (more than 50% for this lake). They also determine a predictive function of oxic methane production contribution depending on littoral sediment area and surface mixed layer volume.

The manuscript is well written. The methodology and different steps are well explained. The authors have well addressed the comments of the first review and they have done a good job in clarifying some missing elements in the methods. As a result, the revised paper will be a significant contribution in the methane literature regarding oxic methane production in lakes. I would recommend publication in Nature Communication.

Reviewer #2 (Remarks to the Author):

This is a careful response to comments. The manuscript has been greatly improved after revision and becomes acceptable.